# Autophagy of OTUD5 destabilizes GPX4 to confer ferroptosis-dependent kidney injury

Li-Kai Chu[1,6], Xu Cao[1,6], Lin Wan[1,6], Qiang Diao[2,6], Yu Zhu[1], Yu Kan[1], Li-Li Ye[1], Yi-Ming Mao[3], Xing-Qiang Dong[1], Qian-Wei Xiong[1], Ming-Cui Fu[1], Ting Zhang[1], Hui-Ting Zhou[1], Shi-Zhong Cai[1], Zhou-Rui Ma[1], Ssu-Wei Hsu[4,5], Reen Wu[5], Ching-Hsien Chen [4,5,7] ✉, Xiang-Ming Yan [1,7] ✉ & Jun Liu [1,7] ✉

Ferroptosis is an iron-dependent programmed cell death associated with severe kidney diseases, linked to decreased glutathione peroxidase 4 (GPX4). However, the spatial distribution of renal GPX4-mediated ferroptosis and the molecular events causing GPX4 reduction during ischemia-reperfusion (I/R) remain largely unknown. Using spatial transcriptomics, we identify that GPX4 is situated at the interface of the inner cortex and outer medulla, a hyperactive ferroptosis site post-I/R injury. We further discover OTU deubiquitinase 5 (OTUD5) as a GPX4-binding protein that confers ferroptosis resistance by stabilizing GPX4. During I/R, ferroptosis is induced by mTORC1-mediated autophagy, causing OTUD5 degradation and subsequent GPX4 decay. Functionally, OTUD5 deletion intensifies renal tubular cell ferroptosis and exacerbates acute kidney injury, while AAV-mediated *OTUD5* delivery mitigates ferroptosis and promotes renal function recovery from I/R injury. Overall, this study highlights a new autophagy-dependent ferroptosis module: hypoxia/ischemia-induced OTUD5 autophagy triggers GPX4 degradation, offering a potential therapeutic avenue for I/R-related kidney diseases.

Acute kidney injury (AKI), a prevalent syndrome characterized by a swift decline in renal function, has progressively emerged as a serious global health problem, with ~13.3 million annual cases and 1.7 million associated deaths[1,2]. AKI can arise from a variety of triggers, including ischemia/reperfusion (I/R) injury, nephrotoxin exposure, and sepsis, with I/R injury implicated in 11–30% of cases[3]. Despite the prominent role of I/R injury in AKI pathogenesis, our current knowledge base of the underlying pathophysiological processes remains incomplete, thereby impeding the advancement of effective therapeutic strategies for AKI. Notably, renal tubular cells, the fundamental structural units of the kidney responsible for pivotal functions such as metabolic processing and reabsorption within the body, are particularly susceptible

to a wide range of deleterious stimuli. This susceptibility leads to a pronounced dysfunction of renal tubular cells, which in turn contributes significantly to the observed pathology in I/R-associated AKI[4]. Consequently, there is an urgent need to gain a comprehensive understanding of the intricate mechanisms driving the initiation and progression of AKI, with a specific focus on renal tubular cell dysfunction.

The onset of ischemia and subsequent reperfusion have profound adverse effects on renal tubular cells. A key consequence is the disruption of intracellular glutathione (GSH) metabolism, leading to excessive accumulation of reactive oxygen species (ROS), closely correlating with tubular cell apoptosis and ferroptosis[5]. Ferroptosis, a

[1]Pediatric Institute of Soochow University, Children's Hospital of Soochow University, Soochow University, 215025 Suzhou, China. [2]Department of Medical Imaging, Jinling Hospital, Affiliated Hospital of Medical School, Nanjing University, 210002 Nanjing, China. [3]Department of Thoracic Surgery, Suzhou Kowloon Hospital, Shanghai Jiao Tong University School of Medicine, 215028 Suzhou, China. [4]Division of Nephrology, Department of Internal Medicine, University of California Davis, Davis, CA, USA. [5]Division of Pulmonary, Critical Care, and Sleep Medicine, Department of Internal Medicine, University of California Davis, Davis, CA, USA. [6]These authors contributed equally: Li-Kai Chu, Xu Cao, Lin Wan, Qiang Diao. [7]These authors jointly supervised this work: Ching-Hsien Chen, Xiang-Ming Yan, Jun Liu. ✉e-mail: jchchen@ucdavis.edu; xm_yan0302@suda.edu.cn; junliu@suda.edu.cn

non-apoptotic form of programmed cell death, arises due to excessive intracellular ROS accumulation, typically resulting from compromised ROS-scavenging systems such as GSH biosynthesis and glutathione peroxidase 4 (GPX4) homeostasis[6]. GPX4 serves as a key inhibitor of ROS-mediated phospholipid peroxidation[7], thus placing it central to ferroptosis regulation and marking it as a potential therapeutic target. The association of renal tubular cell ferroptosis with AKI[8,9] postulates that ferroptosis inhibition could be a novel AKI treatment strategy. Yet, the specific mechanisms maintaining GPX4 homeostasis in renal tubular cells during ferroptosis largely remain undefined.

The maintenance of intracellular GPX4 homeostasis is stringently regulated by the interplay between the ubiquitin–proteasome system (UPS) and deubiquitinating enzymes (DUBs), major regulatory mechanisms for protein homeostasis[10]. Such mechanisms have been reported as crucial in regulating GPX4 homeostasis under various conditions. For instance, the E3 ligase TRIM46 has been shown to facilitate the ubiquitin-mediated proteasomal degradation of GPX4 in endothelial cells[11], while the E3 ligase MIB2 similarly regulates GPX4 in neurons[12]. Additionally, the linear ubiquitin chain assembly complex (LUBAC) stabilizes GPX4 in fibroblasts in response to I/R[13]. Thus, identifying UPS/DUB proteins that target GPX4 for degradation or stabilization presents a viable therapeutic opportunity for ferroptosis-related diseases.

In this study, we explored the role of cell ferroptosis in I/R-associated AKI. We revealed that ferroptosis occurs in renal tubular cells, leading to a reduction in GPX4 via ubiquitin-proteasomal degradation in response to I/R injury. We then identified ovarian tumor domain-containing 5 (OTUD5) as a GPX4-interacting protein that promotes resistance to ferroptosis during I/R by stabilizing GPX4 expression. Additionally, we demonstrated that I/R downregulates OTUD5 through the activation of lysosomal degradation controlled by the mammalian target of rapamycin complex 1 (mTORC1). Overall, our findings indicate the therapeutic potential of an OTUD5-guided, ferroptosis-based approach for the treatment of patients with I/R-associated AKI.

## Results

### Spatial resolution of GPX4-regulated tubular cell ferroptosis in I/R-induced AKI

To elucidate the comprehensive molecular alterations in kidneys exposed to ischemia-reperfusion (I/R), we generated a mouse model of bilateral I/R-induced acute kidney injury (AKI) (Supplementary Fig. 1a, b)[14]. Subsequently, we performed single-cell RNA sequencing (scRNA-seq) on the mouse kidneys. Following quality control and batch effect correction (Supplementary Fig. 1c), 10,971 and 10,398 cells from sham and I/R-treated kidneys were designated for graph-based clustering and UMAP-based dimensionality reduction, respectively (Fig. 1a). Cell clusters were annotated by comparing their transcriptional profiles with known cell-type-specific markers (Supplementary Fig. 1d). Among the 10 identified cell populations, we observed a marked decrease in proximal tubular cells and an upswing in innate immune cells, including macrophages and neutrophils (Fig. 1a and b). Given the pronounced alterations in the proximal tubular cell population, we conducted a gene set enrichment analysis (GSEA) on their differentially expressed genes and noticed that signaling pathways associated with ferroptosis were significantly enriched (Fig. 1c). Next, we executed spatial transcriptomics on the kidney sections (Supplementary Fig. 1e). Notably, kidney injury marker *Lcn*, along with core genes of inflammation and NF-kB signaling, were augmented in I/R-treated kidneys (Supplementary Fig. 1f–h), suggesting tubular cell injury. Consistent with scRNA-seq data, a ferroptosis gene signature was upregulated in the I/R group (Fig. 1d).

We next confirmed the presence of ferroptosis in AKI by finding that I/R-treated kidneys exhibited tubular cell death accompanied by lipid peroxidation, as indicated by TUNEL staining and elevated 4-HNE levels (Fig. 1e, Supplementary Fig. 1i). Moreover, we observed

decreased expression of glutathione peroxidase 4 (GPX4), an enzyme vital for neutralizing phospholipid hydroperoxides (PLOOHs) (Fig. 1f), in protein levels (Fig. 1g). To delineate the cellular distribution of GPX4-mediated tubular cell ferroptosis, we examined the spatial expression pattern of *Gpx4*. Intriguingly, *Gpx4* was predominantly expressed in Cluster 1 of sham-treated kidney and Cluster 3 of I/R-treated kidney (Supplementary Fig. 1j), situated at the interface of the inner cortex and outstrip of the medulla. We observed a minor decrease in *Gpx4 mRNA* expression in this region in I/R-treated kidneys (Fig. 1h). To corroborate our findings ex vivo, we created a hypoxia/reoxygenation (H/R) model using primary renal tubular cells derived from the inner cortex of mouse kidneys[15]. H/R induction triggered a decrease in GPX4 protein levels (Fig. 1g) and induced substantial phospholipid peroxidation (Fig. 1i), as evidenced by Liperfluo staining[16]. Furthermore, ferroptotic cell death was detected using the fluorescent lipid peroxidation sensor BODIPY™ 581/591 C11 and cell death Dye 7-AAD to identify cell ferroptosis (Fig. 1j). Inconsistency with protein reduction, the mRNA level of *Gpx4* was not markedly decreased upon I/R induction (Supplementary Fig. 1k and l). Altogether, our findings suggest the significant role of post-transcriptional mechanisms in the reduction of GPX4 following I/R induction, while the contribution of transcriptional regulation appears to be relatively limited.

### OTUD5 interacts with GPX4 to stabilize it in response to H/R

Our subsequent investigation targeted the molecular events underpinning I/R-induced GPX4 reduction. Two common mechanisms for protein post-transcriptional degradation are ubiquitin-dependent proteasomal degradation and autophagy-dependent lysosomal degradation[17]. We employed the proteasomal inhibitor MG132 and the lysosome inhibitor chloroquine (CQ) to specifically block these degradation pathways. Both were found to counteract the loss of GPX4 triggered by H/R, suggesting their involvement in modulating GPX4 homeostasis during H/R exposure (Fig. 2a). We next confirmed the contribution of these pathways to GPX4 decay induced by H/R. Even though GPX4 was not significantly elevated in the lysosome post-H/R induction (Supplementary Fig. 2a), GPX4 ubiquitination levels showed a notable increase with H/R treatment (Fig. 2b), suggesting a potential primary role for the ubiquitin-dependent degradation pathway in GPX4 reduction. The ubiquitin-proteasome system (UPS) and deubiquitinating enzymes (DUBs) are critical for cellular protein homeostasis, executing both post-transcriptional degradation and stabilization[18]. They also play a crucial role in ferroptosis and kidney disease pathology[10,19]. As expected, many UPS/DUBs were abnormally expressed in I/R-exposed mouse kidneys, as revealed by transcriptome sequencing analysis (Supplementary Fig. 2b). We further uncovered any potential UPS/DUB proteins directly targeting GPX4 for degradation. Mass spectrometry (MS) of the purified GPX4 complex allowed us to identify 267 proteins exclusively co-immunoprecipitating with GPX4, including the deubiquitinating proteins OTUB1 and OTUD5, and the E3 ubiquitin ligases TRIM21, UBR5, and XIAP (Fig. 2c).

Upon investigating the expression of the five identified UPS/DUB proteins, we found that levels of OTUD5, OTUB1, and UBR5 were reduced, while levels of TRIM21 and XIAP remained relatively stable following H/R induction (Supplementary Fig. 2c). This suggested that OTUD5 and OTUB1 are potentially responsible for GPX4 stabilization. To verify this, we performed siRNA experiments targeting OTUB1 and OTUD5. We found that upon H/R induction, GPX4 expression drastically reduced with OTUD5 knockdown, an effect not observed in siOTUB1-infected cells (Fig. 2d), implying that OTUD5 could be the deubiquitinating protein responsible for GPX4 stabilization. Spatial resolution revealed the localization of OTUD5 and GPX4 at the mRNA level (Supplementary Fig. 2d). Subsequent reciprocal immunoprecipitation experiments confirmed endogenous interaction between GPX4 and OTUD5, with their interaction being reduced upon H/R

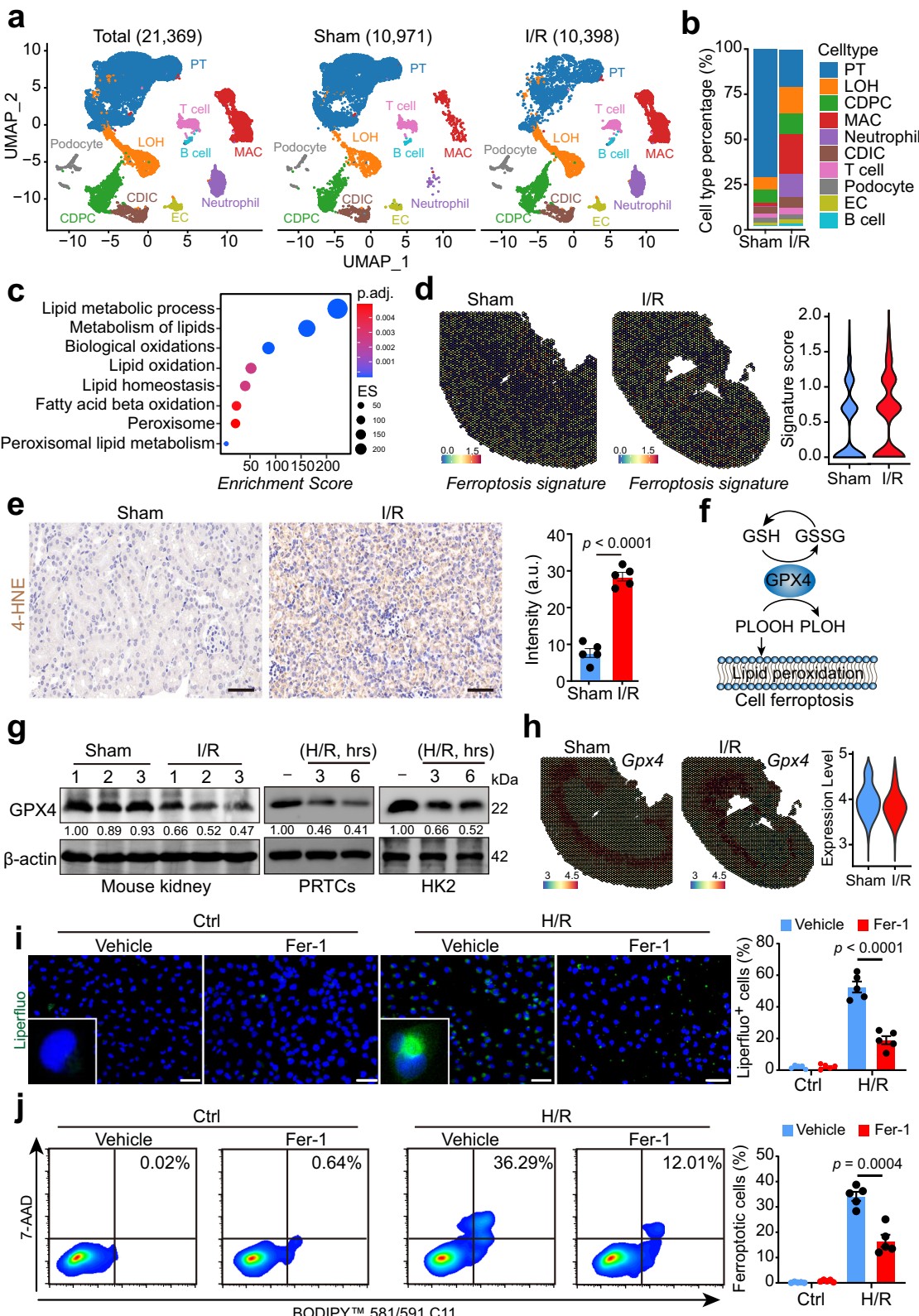

induction (Fig. 2e). Furthermore, immunofluorescence analysis confirmed the endogenous colocalization of OTUD5 and GPX4 in the cytosol. Notably, both the expression levels and colocalization of these proteins were reduced following H/R induction. (Fig. 2f).

We hypothesized that OTUD5 could stabilize GPX4 through its deubiquitinating activity, and that H/R-induced GPX4 reduction results from a deficiency in OTUD5 function. Our siRNA experiments

confirmed this, revealing stable GPX4 expression in OTUD5-deficient cells under physiological conditions. However, the expression levels of GPX4 and OTUD5 were significantly reduced in cells harboring WT OTUD5, and the loss of GPX4 was markedly increased, along with aggravated ubiquitination, in OTUD5-deficient cells upon H/R stimulation (Fig. 2g). To verify the gain-of-function, we depleted endogenous OTUD5 and subsequently transfected a His-tagged plasmid

**Fig. 1 | Spatial resolution of GPX4-mediated tubular cell ferroptosis in I/R-induced AKI.** Wild-type C57BL/6J mice were subjected to bilateral ischemia/reperfusion (I/R) injury surgery, or a sham operation, and were sacrificed 48 h after the surgery. Injured and normal sham kidneys were harvested for experiments, including single-cell-RNA-sequencing and spatial transcriptomics. **a** UMAP projection of 10,971 cells from the sham kidney, and 10,398 cells from the injured kidney; cell identity was annotated based on cell type-specific markers (see the "Methods" section). **b** A bar plot shows the percentage of each cell type out of the total cells in each group. PT proximal tubular, LOH Loop of Henle, CDPC collecting duct principal cell, MAC macrophage, CDIC collecting duct intercalated cell, EC endothelial cell. **c** A dot plot shows the ferroptosis-associated signaling pathways among the enriched signaling pathways, using GSEA analysis based on the scRNA-seq data of PT cells from I/R-treated and sham kidneys. **d** Spatial feature plots and violin plots of the ferroptosis signature score in ST spots. **e** IHC staining and quantification of 4-HNE expression on kidney sections from sham or I/R-treated mice ($n = 5$); scale bars, 50 µm, a.u.: arbitrary units. **f** Schematic of GPX4-mediated lipid peroxidation and cell ferroptosis. **g** Immunoblot and quantification of GPX4 in mouse kidneys ($n = 3$ per group), primary renal tubular cells (PRTCs), and HK2 cells. **h** Spatial feature plots and violin plots of *Gpx4* in ST spots from sham or I/R-treated mouse kidneys. **i** Representative images and quantification of cell membrane lipid peroxidation stained by the liperfluo probe ($n = 5$ independent experiments), scale bars, 50 µm. **j** Ferroptosis was measured using the fluorescent lipid peroxidation sensor BODIPY™ 581/591 C11 and cell death Dye 7-AAD in HK2 cells treated with or without Fer-1 following H/R induction ($n = 5$ independent experiments). Data are presented as mean ± s.e.m.; statistical significance was determined using an unpaired two-tailed Student's *t*-test.

containing WT OTUD5 into OTUD5-depleted cells. We observed that overexpression of OTUD5 did not impact GPX4 expression under physiological conditions but notably prevented H/R-induced proteasomal degradation, thereby promoting GPX4 abundance (Fig. 2h). The enzymatic activity of OTUD5 is crucial for its deubiquitinating function[20,21]. To confirm this enzymatic activity for GPX4 stabilization, we co-transfected GPX4 with either wild-type (WT) OTUD5 or an enzymatically inactive variant of OTUD5 harboring the C224S mutation (C224S) into cells. We found that, unlike WT OTUD5, the mutant OTUD5 (C224S) was unable to stabilize GPX4 (Fig. 2i). We then asked if OTUD5 could prevent GPX4 degradation. We treated cells with cycloheximide (CHX) to block protein synthesis and observed that the GPX4 degradation timeline significantly shortened in OTUD5-depleted cells under H/R induction (Fig. 2j). These results support the notion that OTUD5 can stabilize GPX4 in renal tubular cells under H/R stimulation.

### Otud5 deletion renders kidneys susceptible to I/R injury

To explore the effect of OTUD5 loss-of-function on GPX4-mediated renal tubular cell ferroptosis and renal function upon I/R induction in vivo, we engineered renal tubular epithelial cell-specific *Otud5* knockout mice (Fig. 3a). This was accomplished by crossing *Pax8*-Cre and *Otud5*-flox mice, leading to specific *Otud5* knockout in renal tubular epithelial cells (Supplementary Fig. 3a and b). These *Pax8*^Cre^/*Otud5*^fl/fl^ mice presented no obvious kidney dysfunction physiologically, and their renal *Gpx4* mRNA expression remained analogous to wild-type *Otud5*^fl/fl^ mice (Supplementary Fig. 3c). However, post-I/R induction, *Pax8*^Cre^/*Otud5*^fl/fl^ mice exhibited an amplified reduction of renal GPX4 (Fig. 3b, Supplementary Fig. 3b), accompanied by a marked upregulation of 4-HNE and increased cell death. These mice also displayed severe kidney injury, as indicated by heightened tubular damage and a notable presence of cellular debris in the tubular lumen (Fig. 3b).

Infiltration of macrophages, immune cells known to mediate the inflammatory response and subsequent resolution post I/R injury[22], was also markedly increased in *Pax8*^Cre^/*Otud5*^fl/fl^ mice, as shown by immunofluorescence assay (Fig. 3b). Additionally, these mice displayed exacerbated renal dysfunction, with elevated SCr and BUN levels compared to WT mice (Fig. 3c and d). Consistently, renal *Lcn2* and *Havcr1* mRNA levels were higher in *Pax8*^Cre^/*Otud5*^fl/fl^ kidneys relative to WT *Otud5*^fl/fl^ mice (Fig. 3e and f). Furthermore, the *Pax8*^Cre^/*Otud5*^fl/fl^ kidneys displayed a higher expression of pro-inflammatory genes, specifically *Il6*, and *Tnf*, than their WT counterparts (Fig. 3g and h). Collectively, these data underscore the critical role of OTUD5 in regulating renal tubular cell ferroptosis and kidney function during I/R injury.

### OTUD5 shields renal tubular cells from ferroptosis following H/R injury

To discern whether OTUD5 could mitigate GPX4-mediated cell ferroptosis and enhance cell survival, we knocked down *OTUD5* in HK2 cells using siRNA. By utilizing the fluorescent lipid peroxidation sensor BODIPY™ 581/591 C11 and cell death Dye 7-AAD to detect cell ferroptosis[16], we discovered that individual depletion of *OTUD5* did not instigate cell ferroptosis, but rather sensitized cells to ferroptosis upon H/R exposure (Fig. 4a). Intriguingly, introduction of WT OTUD5, as opposed to catalytically inactive C224S OTUD5, diminished H/R-induced cell ferroptosis (Fig. 4b), indicating a protective role of OTUD5 against renal tubular cell ferroptosis upon H/R. Supporting this assertion, cell viability assays revealed that OTUD5 depletion led to reduced cell viability and slower recovery from H/R (Fig. 4c), whereas the introduction of WT OTUD5, instead of C224S OTUD5, increased cellular resistance to H/R-induced ferroptosis (Fig. 4d). To verify these findings, we isolated PRTCs from the kidneys of *Pax8*^Cre^/*Otud5*^fl/fl^ and their littermate WT *Otud5*^f/f^ mice and subjected them to H/R induction. Liperfluo staining for lipid peroxide revealed that PRTCs from *Pax8*^Cre^/*Otud5*^fl/fl^ mice were more susceptible to H/R-induced peroxidation and ferroptosis compared to those from WT mice (Fig. 4e).

We further investigated whether OTUD5 protected renal tubular cells from H/R-induced ferroptosis by stabilizing GPX4. The use of the GPX4 inhibitor RSL3 triggered physiological cell ferroptosis and eliminated the protective effects of OTUD5 under H/R conditions (Fig. 4f), thus confirming the essential nature of the OTUD5/GPX4 axis in mediating renal tubular cell ferroptosis. Additionally, we assessed whether the protective impact of OTUD5 against ferroptosis was unique to H/R. We discovered that altering OTUD5 levels through overexpression or knockdown did not significantly affect ferroptosis triggered by erastin or RSL3 (Supplementary Fig. 4a–d), indicating that the influence of OTUD5 is indeed specific to H/R conditions. Given that GPX4 modulates cell ferroptosis by altering GSH metabolism[23], we tested whether OTUD5 also reprogrammed GSH metabolism. We observed that both the total intracellular GSH level and GSH/GSSG ratio (an indicator of intracellular antioxidative ability) declined following H/R and this decline was more pronounced in cells lacking OTUD5 (Fig. 4g). Conversely, OTUD5 overexpression mitigated the antioxidant disability observed upon H/R induction (Fig. 4h). These findings suggest that OTUD5, through GPX4 stabilization, protects renal tubular cells from I/R-induced ferroptosis.

### Hypoxia triggers autophagic degradation of OTUD5

Given the above observations, we sought to investigate the reason behind the reduction of OTUD5 following H/R induction. We initially defined the spatial expression pattern of *Otud5*, noting its consistent presence across the whole renal parenchyma, except for a reduction in Cluster 4 of the sham kidney, identified as the renal pelvis (Supplementary Fig. 5a). And, qPT-PCR analysis confirmed no significant changes were observed in *Otud5* expression after H/R treatment (Supplementary Fig. 5b), hinting at non-transcriptional mechanisms contributing to OTUD5 reduction. To elucidate the underlying mechanisms, we conducted RNA sequencing (RNA-seq) on H/R-treated PRTCs. KEGG analysis indicated that H/R activated autophagy and several well-known signaling pathways, including the hypoxia-related

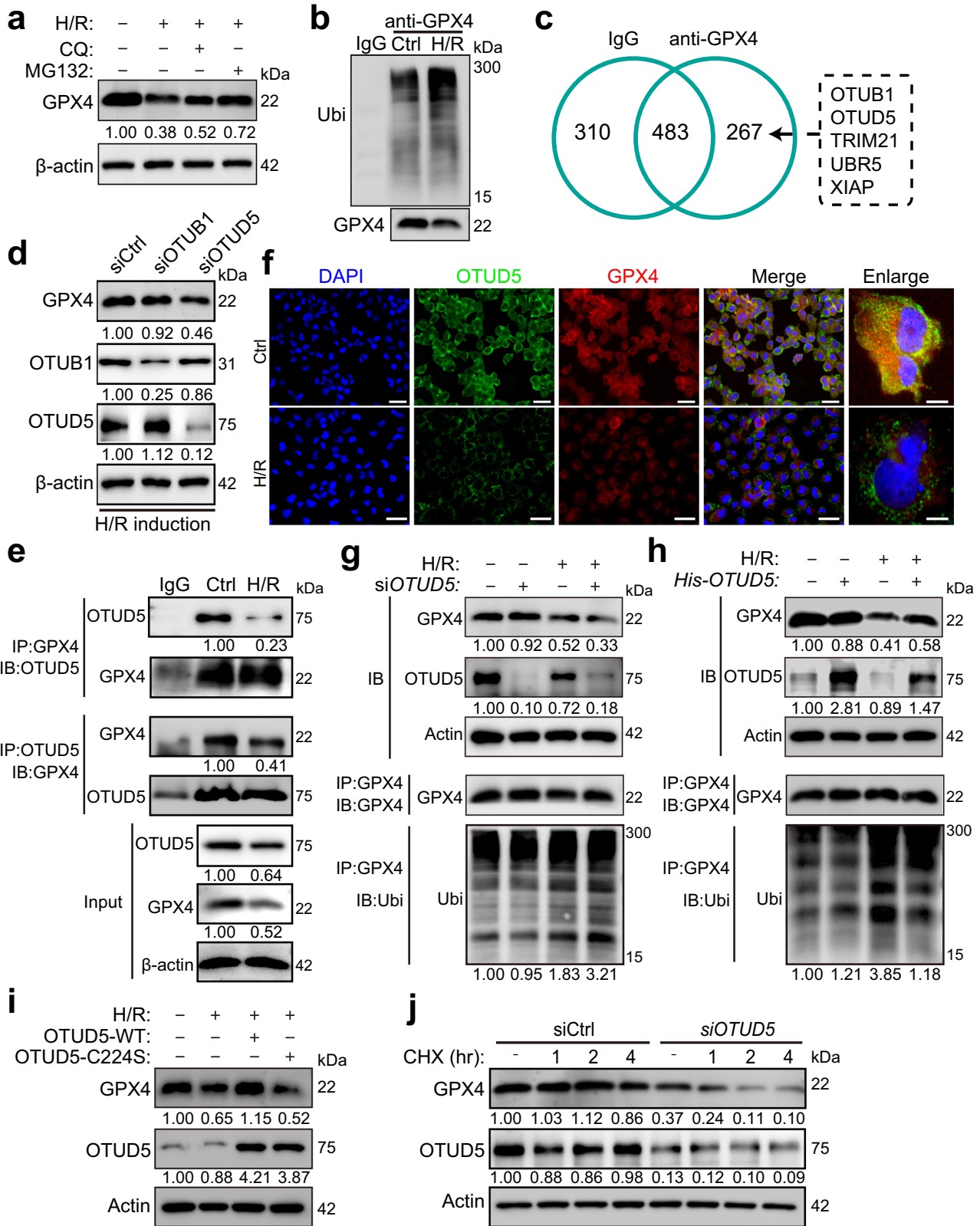

HIF-1 signaling, mTOR activity[24], and PI3K/AKT[25] and MAPK[26] linked to autophagy regulation (Fig. 5a). GSEA analysis of scRNA-seq data also revealed enrichment of cellular response to hypoxia and autophagy pathways in PT cells of I/R-treated kidneys (Fig. 5b). Likewise, the spatial transcriptomic analysis demonstrated an increased expression of autophagy gene signature in injured kidneys compared to sham ones (Fig. 5c). Notably, we observed a parallelism between enhanced

autophagy signature expression and decreased *Otud5* expression following I/R injury (Fig. 5d).

Autophagy, a lysosome-dependent protein degradation mechanism crucial for cellular homeostasis[27], emerged as a potential mediator of OTUD5 reduction following H/R. Initial evidence showed substantial autophagy activation under H/R conditions (Fig. 5e and f). Importantly, the use of the lysosome inhibitor chloroquine (CQ) effectively

**Fig. 2 | OTUD5 is a GPX4-interacting protein for stabilization in response to I/R.**
**a** PRTCs were induced by H/R in the presence or absence of the proteasome inhibitor MG132, or the lysosome inhibitor chloroquine (CQ) for 3 h. Cells were collected, and the protein levels were analyzed by immunoblotting. **b** Immunoblot and quantification analysis of GPX4's ubiquitination in PRTCs upon H/R induction. **c** The protein lysate of HK2 cells was combined with an anti-human GPX4 antibody or the isotype IgG for immunoprecipitation. The number of GPX4 interacting proteins in the protein complex was identified using LC−MS/MS. **d** Immunoblot and quantification analysis of GPX4 expression in HK2 cells infected with siOTUB1 or siOTUD5, respectively. **e** HK2 cells were treated with H/R for 3 h, and the cell lysates were mixed with an anti-human GPX4 or anti-human OTUD5 antibody for immunoprecipitation. The immunoprecipitated protein complex was collected, and protein levels were detected by immunoblotting. **f** Representative fluorescence images of GPX4 and OTUD5 expression in cells treated with or without H/R for 3 h. Scale bar = 50 or 10 μm. **g** Immunoblotting and quantification of GPX4 and OTUD5 expression, and GPX4's ubiquitination in siOTUD5-infected HK2 cells treated with or without H/R for 3 h. **h** Immunoblotting and quantification of GPX4, OTUD5, and GPX4's ubiquitination in His-tagged OTUD5 plasmid-infected HK2 cells treated with or without H/R for 3 h. **i** Immunoblotting and quantification of GPX4 and OTUD5 expression in HK2 cells infected with WT OTUD5 or C224S OTUD5 and subjected to H/R for 3 h. **j** Immunoblotting and quantification of GPX4 and OTUD5 expression in siOTUD5-infected HK2 cells, which were pretreated with Cycloheximide (CHX) for 1, 2, or 4 h and subjected to H/R for 3 h.

counteracted H/R-induced OTUD5 reduction and contributed to GPX4 abundance (Fig. 5g). Of the three main autophagy pathways, macroautophagy, endosomal microautophagy, and chaperone-mediated autophagy (CMA)[28], only inhibition of macroautophagy via the inhibitor 3-methyladenine (3-MA) rescued OTUD5 and GPX4 expression (Fig. 5g). Using siRNAs targeting HSC70 and VPS4AB, respectively, neither the inhibition of CMA nor endosomal microautophagy showed any effects on OTUD5 and GPX4 expression (Supplementary Fig. 5c and d). Further validating these findings, we revealed that genetic knockdown of autophagy regulator ATG5 prevented H/R-induced OTUD5 and GPX4 loss (Fig. 5h). Immunofluorescence analysis highlighted the H/R-induced enhancement of autophagy-executing protein LC3 expression and its colocalization with OTUD5 (Fig. 5i). An immunoprecipitation assay showed an increased interaction between LC3 and OTUD5 under H/R conditions (Fig. 5j).

We subsequently examined whether autophagy could bypass OTUD5 to degrade GPX4 directly. Surprisingly, we observed that H/R did not augment the lysosomal level of GPX4 (Supplementary Fig. 2a), and 3-MA failed to rescue H/R-induced GPX4 reduction in OTUD5-depleted cells (Supplementary Fig. 5e). This indicates that OTUD5 plays an essential role in stabilizing GPX4. Considering other potential post-transcriptional mechanisms, we investigated the ubiquitin ligase UBR5, which has been reported to mediate OTUD5 abundance post-transcriptionally[29]. Our prior data also demonstrated potential interaction between UBR5 and OTUD5, as both were part of the GPX4 complex (Fig. 2c). However, UBR5 knockdown failed to rescue H/R-induced OTUD5 reduction (Supplementary Fig. 5f), ruling out the potential role of ubiquitin degradation in the reduction of OTUD5 in response to H/R. Taken together, our results strongly suggest that H/R induces the autophagic degradation of OTUD5.

### H/R diminishes OTUD5 by suppressing mTORC1 signaling
Autophagy, which is activated when mTOR signaling is inhibited, allows cells to survive under stress[24]. We aimed to investigate whether mTORC1 activity regulates autophagic degradation of OTUD5 induced by H/R and whether it protects cells from GPX4-mediated ferroptosis. Through the lens of spatial transcriptomics applied to injured renal tissues, we found an enhanced expression of the mTOR activity signature (Fig. 6a), in line with our RNA-seq data (Fig. 5a). We also observed that H/R induced a time-dependent activation of mTORC1 activity (Fig. 6b). Instead of the mTORC1 inhibitor rapamycin, the mTORC1 activator MHY1485 significantly restored the levels of OTUD5 and GPX4 reduced by H/R (Fig. 6c), suggesting that mTORC1 plays a beneficial role in maintaining OTUD5 homeostasis. During ischemia, renal mTORC1 activity is known to be repressed, a response that is reversed during the reperfusion stage[30]. We analyzed mTOR activity during the hypoxic phase (oxygen deprivation) to investigate whether hypoxia suppresses mTORC1 activity. We utilized HK2 cells, which possess a higher endogenous mTOR activity compared to primary renal tubular cells (Supplementary Fig. 6a). Our findings indicated that hypoxia inhibited mTORC1 activity as well as the protein levels of OTUD5 and GPX4 (Fig. 6d), suggesting that H/R-induced OTUD5

reduction occurs at the early stage of hypoxia. Hypoxia-induced mTOR inhibition is dependent on the negative regulator TSC1[31]. Consistent with this, we found that the knockdown of TSC1 substantially protected OTUD5 and GPX4 from hypoxia-induced loss (Fig. 6e). Further, we explored the functional role of mTORC1 signaling in GPX4-mediated renal tubular cell ferroptosis. Our data showed that MHY1485 significantly mitigated cellular oxidation status (Supplementary Fig. 6b and c), autophagy (Fig. 6f), and cell ferroptosis (Fig. 6g) in response to hypoxia. Overall, these findings suggest that mTORC1 regulates GPX4 homeostasis via OTUD5 autophagy, which plays a vital role in enabling renal tubular cells to resist I/R-induced ferroptosis.

### AAV-mediated OTUD5 therapy protects renal function against I/R injury
Given our findings, we proceeded to investigate whether OTUD5 could serve as a potential therapeutic target for I/R-associated AKI. We employed an intravenous injection strategy of mouse *Otud5*-packaged AAV to create a mouse strain that consistently expresses OTUD5 (Fig. 7a). Mice receiving AAV-delivered *Otud5*, henceforth referred to as OTUD5+ mice, successfully demonstrated ectopic expression of OTUD5 (Supplementary Fig. 7b). These OTUD5+ mice displayed overall normality with similar GPX4 levels at both mRNA and protein levels when compared to their littermate WT counterparts (Supplementary Fig. 7b). When subjected to I/R injury, the kidneys of OTUD5+ mice showed a minor upregulation of mTORC1 activity and a decrease in autophagy compared to WT mice (Supplementary Fig. 7c), suggesting the existence of a potential feedback suppression mechanism. Importantly, the widespread degradation of GPX4 was noticeably less, and GPX4 levels were higher in OTUD5+ mice compared to WT mice (Supplementary Fig. 7c). In terms of functionality, the kidneys of OTUD5+ mice presented with lower levels of BUN, SCr, and lipid peroxidation (as indicated by 4-HNE expression), as well as reduced cell death and immune infiltration (Fig. 7b–d). Specifically, the injured kidneys of OTUD5+ mice exhibited reduced levels of kidney injury biomarkers such as *Lcn*, *Havcr1*, and macrophage infiltration (Fig. 7e–g) as well as pro-inflammatory genes *Il6*, and *Tnf* compared to WT mice (Fig. 7h, i). Taken as a whole, our findings highlight the role of OTUD5 in preserving renal functionality during I/R injury.

### Discussion
Kidney exposure to I/R triggers a series of detrimental events including apoptosis, ferroptosis, and necroptosis within renal tubular cells. These events can transition into AKI and may further progress to irreversible chronic kidney diseases (CKD) and kidney fibrosis[32]. Given that ferroptosis is a key molecular event contributing to AKI, its inhibition emerges as a promising strategy for AKI treatment. Our current study elucidates the spatial distribution and regulatory mechanisms of GPX4-mediated tubular cell ferroptosis in the context of I/R-induced AKI. We show that I/R instigates ferroptosis in renal tubular cells by suppressing the protein level of GPX4 through post-transcriptional modifications and identify the deubiquitinase protein OTUD5 as a

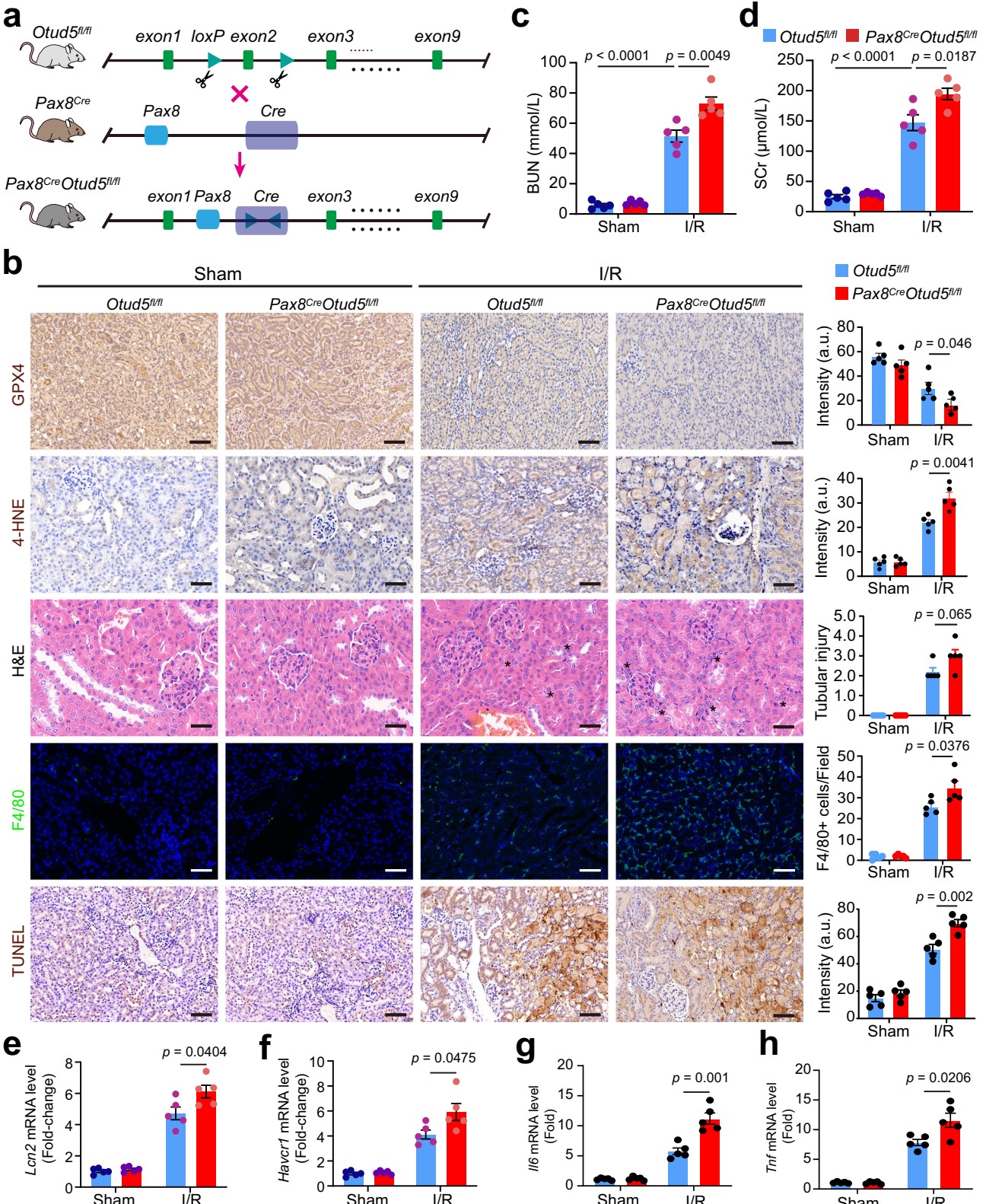

**Fig. 3 | Deletion of Otud5 renders kidneys vulnerable to I/R. a** The schematic shows the crossing of *Pax8*-Cre mice and *Otud5*-floxed mice to generate renal tubular cell conditional *Otud5* knockout (*Pax8^{Cre}Otud5^{fl/fl}*) mice. **b** 4–6-week-old *Pax8^{Cre}Otud5^{fl/fl}* mice and their WT littermates (*n* = 5) were subjected to kidney I/R surgery. After 48 h, kidneys were collected and subjected to H&E, IHC staining, immunofluorescence analysis, TUNEL staining and quantification for kidney injury score evaluation, GPX4 and 4-HNE expression, F4/80 expression, and cell death, respectively; Scale bars, 50 μm. The asterisk indicates the injured tubular area. a.u.: arbitrary units. **c**–**h** Bar plots show the BUN (**c**), SCr levels (**d**), and mRNA level of kidney injury markers *Lcn* (**e**) *Il6* (**g**), and *Tnf* (**h**) of *Pax8^{Cre}Otud5^{fl/fl}* mice and their WT littermates in response to I/R. All values are presented as mean ± s.e.m., *n* = 5; *p* values were calculated by unpaired two-tailed Student's *t*-test.

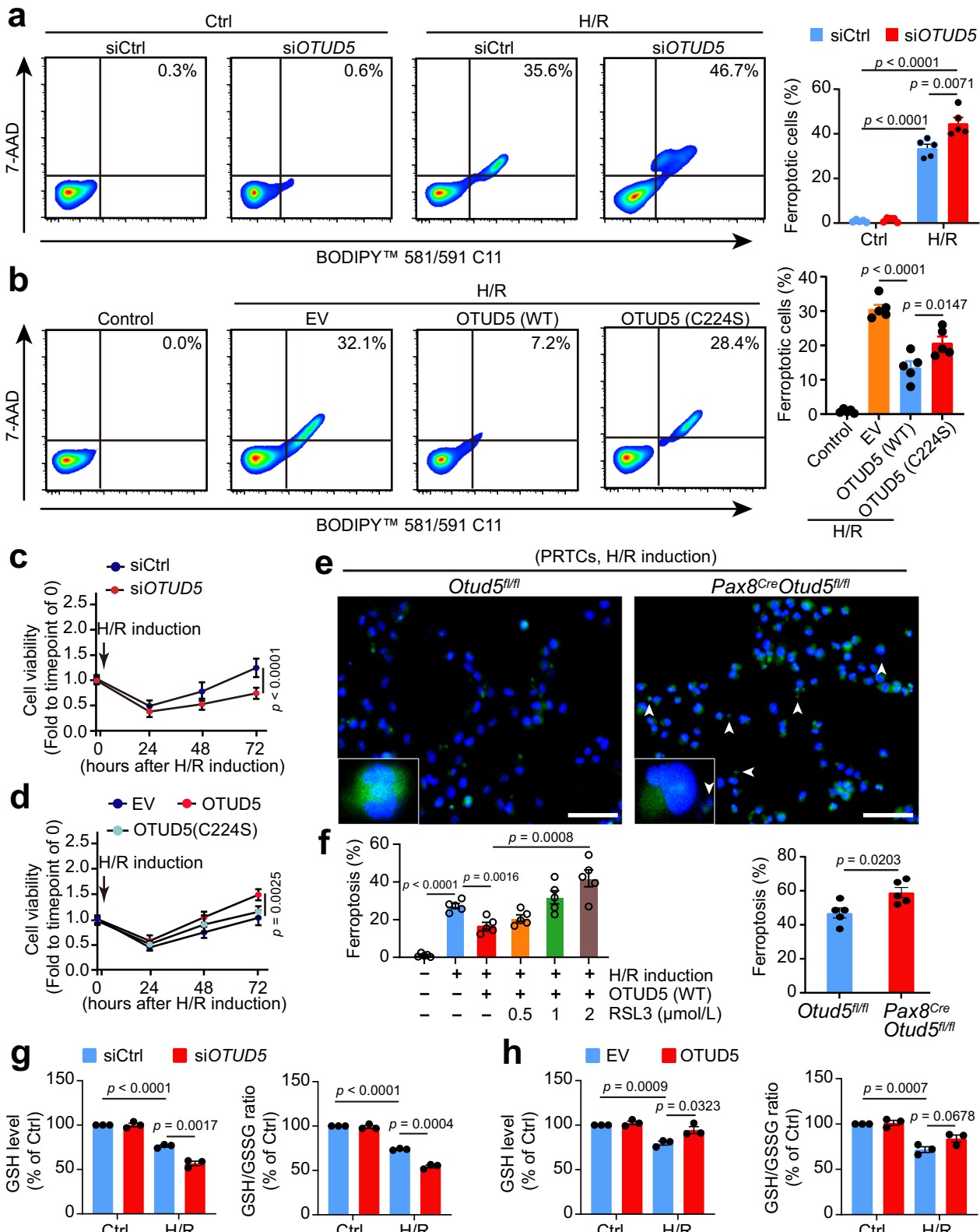

protective partner of GPX4, shielding this protein from ubiquitin proteasomal degradation. We reveal that OTUD5 confers resistance to ferroptosis in response to I/R in renal tubular cells in vitro and mitigates the severity of AKI in vivo. From a mechanistic perspective, our study provides evidence that the ubiquitin-mediated degradation of GPX4 induced by I/R is due to the compromised ability of OTUD5 to deubiquitinate GPX4, a process compromised by hypoxia induction.

Furthermore, our research unravels the mechanism regulating the reduction of OTUD5. Specifically, hypoxia inhibits mTOR activity, thereby activating the autophagy lysosomal pathway, which subsequently targets OTUD5 for degradation. Our investigation provides the first evidence of a mTORC1/OTUD5/GPX4 regulatory axis in renal tubular cells under I/R conditions. This regulatory axis presents a promising target for therapeutic intervention in I/R-associated AKI.

**Fig. 4 | OTUD5 protects renal tubular cells from ferroptosis in response to H/R injury. a** Ferroptosis was measured using fluorescent lipid peroxidation sensor BODIPY™ 581/591 C11 and cell death Dye 7AAD in siControl or si*OTUD5*-transfected cells 3 h after H/R induction (*n* = 5 independent experiments). **b** Ferroptosis was measured in empty vector (EV), WT *OTUD5*, or enzymatically inactive *OTUD5* (C224S) plasmid-transfected HK2 cells after H/R induction (*n* = 5 independent experiments). **c** Cell viability was measured in siControl or si*OTUD5*-transfected cells at 24, 48, and 72 h after ex vivo I/R induction (*n* = 5 independent experiments). **d** Cell viability was measured in EV, WT *OTUD5*, or *OTUD5*-C224S-transfected cells after H/R induction (*n* = 5 independent experiments). **e** Representative images and quantification of PRTCs ferroptosis from WT and *Pax8^Cre^Otud5^fl/fl^* mouse stained by liperfluo; white arrowheads represent the broken nucleus in ferroptotic cells (*n* = 5 independent experiments), scale bars: 50 μm. The arrowhead indicates the broken nucleus. **f** Cell ferroptosis was measured in WT OTUD5-transfected cells, either in the presence or absence of various doses of the GPX4 inhibitor RSL3, under H/R induction (*n* = 5 independent experiments). **g** GSH/GSSG ratio and intracellular GSH level were measured in siControl or si*OTUD5*-transfected cells after H/R induction (*n* = 3 independent experiments). **h** GSH/GSSG ratio and intracellular GSH level were measured in EV or WT *OTUD5* plasmid-transfected cells after ex vivo I/R induction (*n* = 3 independent experiments). Data are presented as mean ± s.e.m.; all statistical significance between groups as indicated was determined using an unpaired two-tailed Student's *t*-test.

The findings from our scRNA-seq and spatial transcriptomics delineate a novel facet of AKI pathophysiology, characterized by the infiltration of pro-inflammatory immune cells alongside the loss of renal tubular cells. Notably, we observed that immune cells, especially macrophages, were overrepresented in the scRNA-seq data compared to histological changes. This discrepancy is likely due to the selective process during the suspension preparation phase, which aims to obtain the necessary cell numbers and viability for sequencing by excluding dead renal tubular cells. Furthermore, there was a marked enrichment of ferroptosis-regulating pathways that became pronounced upon I/R insult. This provides compelling evidence for the understanding that ferroptosis, an iron-dependent cell death module, is a major player in AKI pathogenesis. Our data are consistent with previous reports that have alluded to the involvement of ferroptosis in AKI[33,34], yet extend them by uncovering the spatial distribution of key enzyme GPX4, which is integral to the regulation of this cell death process. We found that *Gpx4*, key in the orchestration of ferroptotic processes, is predominantly expressed in a region highly susceptible to I/R injury, i.e., the interface of the inner cortex and the medulla. This spatial resolution adds a new dimension to our comprehension of the heterogeneity of renal susceptibility to I/R injury. While our observation of a decrease in GPX4 expression upon I/R insult aligns with the known role of GPX4 in preventing ferroptosis[35], it provides novel insights into its spatial distribution, thereby enriching the current knowledge base on the tissue-level regulation of ferroptosis in AKI.

Our investigation illuminates key insights into the molecular mechanisms underlying I/R-induced GPX4 reduction, contributing significantly to the broader knowledge base of GPX4 regulation and its role in AKI. We observed that the ubiquitin-dependent proteasomal degradation pathway, mediated by the deubiquitinating enzyme OTUD5, plays a central role in the stability of GPX4, contrasting with previous studies that suggested an almost exclusive role of lysosomal degradation in post-transcriptional regulation of GPX4[36]. This discrepancy highlights the complexity of the post-transcriptional regulation of GPX4 and emphasizes the need for further studies. In agreement with prior research underscoring the role of the ubiquitin-proteasome system (UPS) in protein homeostasis[19], our work further extends this knowledge by identifying OTUD5 as a specific deubiquitinating enzyme involved in GPX4 regulation. The identification of OTUD5 as a critical stabilizer of GPX4 presents an intriguing link between UPS and ferroptosis, areas that have traditionally been studied separately. Our work thereby sets the stage for further examination of OTUD5 and the UPS in regulating GPX4, with direct implications for understanding cell ferroptosis and AKI. Extension of our findings could potentially elucidate novel therapeutic strategies for AKI. Specifically, the role of OTUD5 in stabilizing GPX4 suggests that therapeutic modulation of its activity might be a feasible strategy for controlling GPX4 levels, thus manipulating the course of ferroptosis and ultimately, the progression of AKI.

Through a series of experiments, we demonstrate that OTUD5 plays a protective role against ferroptosis and contributes to cell survival. The protective role of OTUD5 is largely linked to its stabilizing effect on GPX4. Furthermore, the severe kidney injury, marked

increase in inflammation, and heightened renal dysfunction exhibited by OTUD5-deleted mice confirm the critical role of OTUD5 in mitigating the damaging effects of I/R injury. This discovery offers a new perspective on the potential function of OTUD5 in modulating inflammation during AKI. As our results demonstrate that H/R conditions cause the autophagic degradation of OTUD5, it opens up the possibility for future research to explore strategies aimed at stabilizing OTUD5 to preserve GPX4 and prevent ferroptosis. Moreover, we found that H/R conditions diminish OTUD5 by suppressing mTORC1 signaling, which creates an intriguing connection between mTORC1 activity and the regulation of ferroptosis. This relationship provides a novel potential research path: the manipulation of mTORC1 signaling as a means to prevent the autophagic degradation of OTUD5. Such investigations could significantly enhance our comprehension of the underlying mechanisms of cell ferroptosis-regulated AKI and could potentially lead to new therapeutic interventions.

Given that OTUD5 destabilization results in GPX4 decay after I/R induction, the molecular mechanisms that control OTUD5 reduction in these conditions have come into focus. OTUD5 is known to be regulated post-transcriptionally by UBR5[29], and also through phosphorylation mediated by mTORC1 signaling[37]. The mTOR signaling pathway is key to maintaining renal tubular homeostasis. Yet, its dysregulation can lead to renal cell death and a variety of kidney diseases, including AKI[38]. Interestingly, rapamycin, an inhibitor of mTORC1, has been observed to impair tubular cell regeneration and delay renal function recovery following AKI[39]. Conversely, it has also been shown to mitigate I/R injury[40]. This illustrates the paradoxical role of mTOR signaling in renal injury, underscoring the need for further elucidation. Our study presents an unexpected and previously unknown mechanism responsible for OTUD5 degradation. We found that the autophagic lysosomal pathway, activated by mTORC1 inhibition under hypoxic conditions, plays a critical role. Prior research has demonstrated that OTUD5 modulates mTORC1 activity by destabilizing its inhibitory core proteins[41]. However, in our current investigation, we did not observe this reciprocal regulation of OTUD5 on mTORC1 activity in renal tubular cells during I/R injury.

Therapeutic strategies targeting mTOR signaling for AKI have been proposed, yielding conflicting outcomes due to the complexity of mTOR's role in renal injury[39,40]. Our findings may shed light on this issue. We discovered that hypoxia suppresses mTOR activity, thereby activating mTOR-mediated lysosomal degradation of OTUD5. This results in a diminished expression of OTUD5, a situation that cannot be rectified during the reperfusion stage. Future research will explore the potential benefits of combined targeting of mTOR and OTUD5 for AKI treatment. On the whole, our study compellingly highlights the critical role of the mTOR/OTUD5/GPX4 signaling pathway in regulating renal tubular cell ferroptosis during I/R. These findings further suggest that targeting this pathway could be an effective approach to treating ferroptosis-associated diseases such as AKI.

The process of I/R involves an initial stage of ischemia and hypoxia, which then transitions into a reperfusion phase involving the restoration of blood flow. The most significant and detrimental damage from I/R is primarily inflicted during the reperfusion stage, with wide-ranging

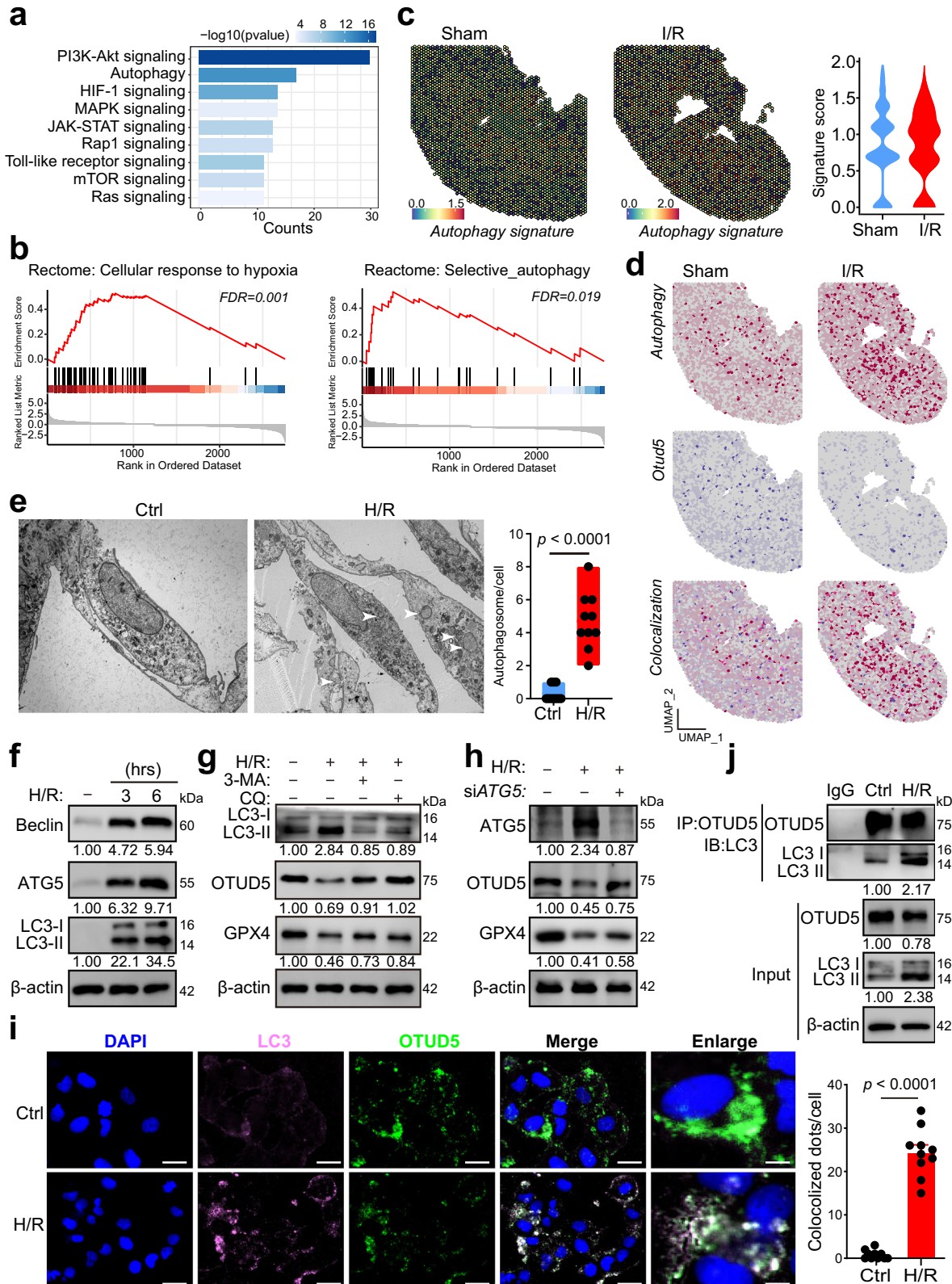

effects including metabolic reprogramming, DNA damage, upregulation of pro-inflammatory genes, and mitochondrial dysfunction[5]. Mitochondrial dysfunction, in particular, can lead to an overload of intracellular ROS and cellular lipid peroxidation. The latter is recognized as a key driver of cellular ferroptosis, a form of regulated cell death[6,7]. In keeping with these insights, our research confirmed that exposure to I/R amplified lipid peroxidation and diminished the expression of GPX4, a pivotal

enzyme in limiting oxidative damage. In addition, our findings revealed the involvement of ferroptosis in modulating the progression of AKI. This was evidenced by the significant reduction in I/R-induced elevated levels of AKI markers, such as Scr, BUN, and kidney injury molecule (KIM), following the administration of the ferroptosis inhibitor, Fer-1[42]. These results significantly contribute to the field of ferroptosis in AKI and suggest potential therapeutic strategies.

**Fig. 5 | Hypoxia activates autophagy degradation of OTUD5. a** PRTCs were treated with H/R for 3 h, followed by RNA isolation and sequencing. A bar plot shows autophagy-associated signaling pathways among the enriched signaling pathways using KEGG analysis. Statistical significance between groups, as indicated, was determined using Fisher's exact test. **b** GSEA analysis shows that the Reactome signaling terms: selective autophagy and cellular response to hypoxia were significantly enriched based on the scRNA-seq data of PT cells from I/R-treated kidneys and sham kidneys. **c** Spatial feature plot and Violin plot of autophagy signature score in ST spots of the two groups. **d** Feature plot of *Otud5* expression, autophagy signature, and their colocalization in ST spots of the two groups. **e** PRTCs were treated with H/R for 3 h and observed for autophagosome formation using a transmission electron microscope (TEM). The representative TEM images and quantification of autophagosomes (white arrowhead) are displayed (*n* = 10

independent cells). *p* values were calculated by unpaired two-tailed Student's *t*-test. **f** PRTCs were treated with H/R for 3 or 6 h, and cells were collected and analyzed by immunoblotting. **g** PRTCs were treated with H/R for 3 h in the presence or absence of the proteasome inhibitor 3-MA, or the lysosome inhibitor chloroquine (CQ). Cells were collected and analyzed by immunoblotting. **h** PRTCs were transfected with siControl or si*ATG5* for 48 h and treated with or without H/R. Cells were collected and analyzed by immunoblotting. **i** Representative fluorescence images of LC3 and OTUD5 in HK2 cells treated with or without H/R (*n* = 10 independent cells). Scale bar = 50 or 10 μm. **j** Cells were treated with H/R for 3 h and collected to analyze the interaction of LC3 and OTUD5 using immunoprecipitation (IP). Data are from three independent experiments and presented as mean ± s.e.m., statistical significance between groups as indicated was determined using an unpaired two-tailed Student's *t*-test.

Previous studies have demonstrated that targeting ferroptosis-regulating genes, such as GPX4, ACSL4, and FSP1 protected renal function against I/R injury[33,43]. However, the efficacy of such strategies is often limited, as the mechanisms regulating ferroptosis are intricate and not fully understood. In an attempt to unravel these complexities, our study unveils a novel ferroptosis-regulating mechanism where OTUD5 is identified as a partner protein that stabilizes GPX4 during I/R injury. We not only decipher the functional role of OTUD5 in mitigating renal tubular cell ferroptosis, but we also explore its protective effects on renal function after I/R injury. Remarkably, the experiments involving AAV-mediated OTUD5 therapy suggest that OTUD5 plays a substantial role in modulating GPX4 regulation and subsequent cell ferroptosis in I/R-induced AKI. By identifying OTUD5 as a potential therapeutic target for AKI associated with I/R, we offer a new avenue for future research into treatments for AKI. The development of a therapeutic strategy that enhances OTUD5 function or expression could potentially ameliorate the deleterious effects of I/R injury on renal function. In light of minor upregulation of mTORC1 activity and a decrease in autophagy in OTUD5⁺ mice, future investigations could also explore the possibility of a feedback suppression mechanism, further elucidating the regulatory complexities in ferroptosis and AKI.

Given the pathological importance of ferroptosis in AKI, our findings have notable implications for the development of future therapeutic strategies. The revelation of GPX4 downregulation in the context of I/R insult points towards a potential therapeutic target. In addition, we pinpoint a region particularly prone to I/R injury marked by heightened ferroptosis and autophagy, offering valuable insight into the pathology of AKI. In the future, building upon these findings will be crucial in decoding the complexities of cell ferroptosis-regulated AKI. In summary, our research enhances our comprehension of ferroptosis's role in AKI, paving the way for future exploration of spatial ferroptotic processes within the kidney. Above all, this study sets a strong foundation for potential intervention strategies by shedding light on a novel autophagy-dependent ferroptosis pathway, presenting possible new directions for treatments in managing I/R-associated kidney diseases.

## Methods
### Mice and mouse model of Ischemia/reperfusion (I/R)-induced AKI
*Otud5*-floxed mice (Strain No. T052115, GemPharmatech, Nanjing, China), and *Pax8*-Cre mice (Strain No. NM-KI-200151, Shanghai Model Organisms Center, Inc., Shanghai, China) were generated using the CRISPR-Cas9 system. For *Otud5*-floxed mice, the guide RNA (gRNA) sequences were designed for regions on exon 2, as illustrated in Fig. 3A. For *Pax8*-Cre mice, a fragment of Cre-WPRE-polyA was inserted into the initiation codon of the Pax8 gene through homologous recombination. The Cas9 mRNA and gRNA were in vitro transcribed and microinjected into the fertilized eggs of C57BL/6J mice. These eggs were then transplanted into a recipient to obtain positive F0 mice, which were then mated with WT C57BL/6J mice to produce positive F1

generation mice. The *Otud5*-floxed mice and *Pax8*-Cre mice were crossed to generate Otud5^fl/fl^Pax8^Cre^ mice, with the Otud5^fl/fl^ mice used as a wild-type control. All mice were housed in a pathogen-free environment, maintained at a temperature of 22 °C, with a 12 h/12 h light/dark cycle, and relative humidity of 50–60%. This study received approval from the Institutional Review Board of the Children's Hospital of Soochow University. Additionally, for the I/R-induced AKI experiments, all procedures were conducted following the guidelines of the Institutional Animal Care and Use Committee (IACUC) of the Children's Hospital of Soochow University. Briefly, 4–6-week-old C57BL/6J mice were intraperitoneally anesthetized with 1% (v/w) chloral hydrate at a dose of 0.8 mg/kg. For ischemia induction, the renal pedicles were clamped with a clip (JY17100-30, JinYan Medical Device Manufacturing Co., Ltd, Zhejiang, China) for 45 min through the incision at the flank and were then followed by reperfusion by clip removal. Incisions were sutured under sterile surgical conditions, and the mice were transferred to the standard housing environment for recovery. After 48 h, mice were placed in a chamber filled with 100% CO₂ for 3 min to induce rapid unconsciousness, followed by cervical dislocation. Under anesthesia, mouse kidneys and blood samples were collected and subjected to analysis, including serum creatinine (Scr), blood urea nitrogen (BUN), and histopathological examination.

### Cell culture, H/R induction ex vivo, and cell viability detection
The normal human proximal tubular cell line, HK2 (catalog number: CRL-2190), and human embryonic kidney cell line, HEK293T (catalog number: CRL-3216), were obtained from the American Type Culture Collection (ATCC) and cultured in DMDM/F12 medium (Gibco) supplemented with 10% fetal bovine serum (FBS, Gibco) and 1% penicillin/streptomycin (Thermo Fisher Scientific). The cells were maintained in a humidified incubator with 5% CO₂ at 37 °C. Mouse primary renal tubular cells (PRTCs) were isolated from both male and female mice, following the method described in a previous study[44]. Briefly, the outer strips of the cortex and the medulla from mouse kidneys were removed with scissors, and the inner cortex was collected, minced, and digested using 1 mg/ml collagenase IV (Sigma-Aldrich) for 40 min, before filtration through a 70 μm strainer. The suspension containing renal tubules was collected and placed in a six-well cell culture plate for cultivation in DMDM/F12 medium (Gibco) supplemented with 10% FBS (Gibco) and 1% penicillin/streptomycin (Thermo Fisher Scientific). The cells were then maintained in a humidified incubator with 5% CO₂ at 37 °C. To mimic I/R induction ex vivo, the cells were cultured under oxygen- and serum-free conditions for 6 h, followed by the re-introduction of normoxia and complete cell culture medium for varying periods as required. This was conducted using a Hypoxia Chamber (MIC-101, Billups Rothenberg Inc.), following the protocols of previous studies[45]. For cell viability detection, the cells were seeded in a 96-well plate, with ~10,000 cells per well, and left overnight. After hypoxia induction, the cell medium was removed and replaced by a fresh medium containing the CCK8 reagent (96992, Sigma Aldrich) following the manufacturer's instructions. The plate was then

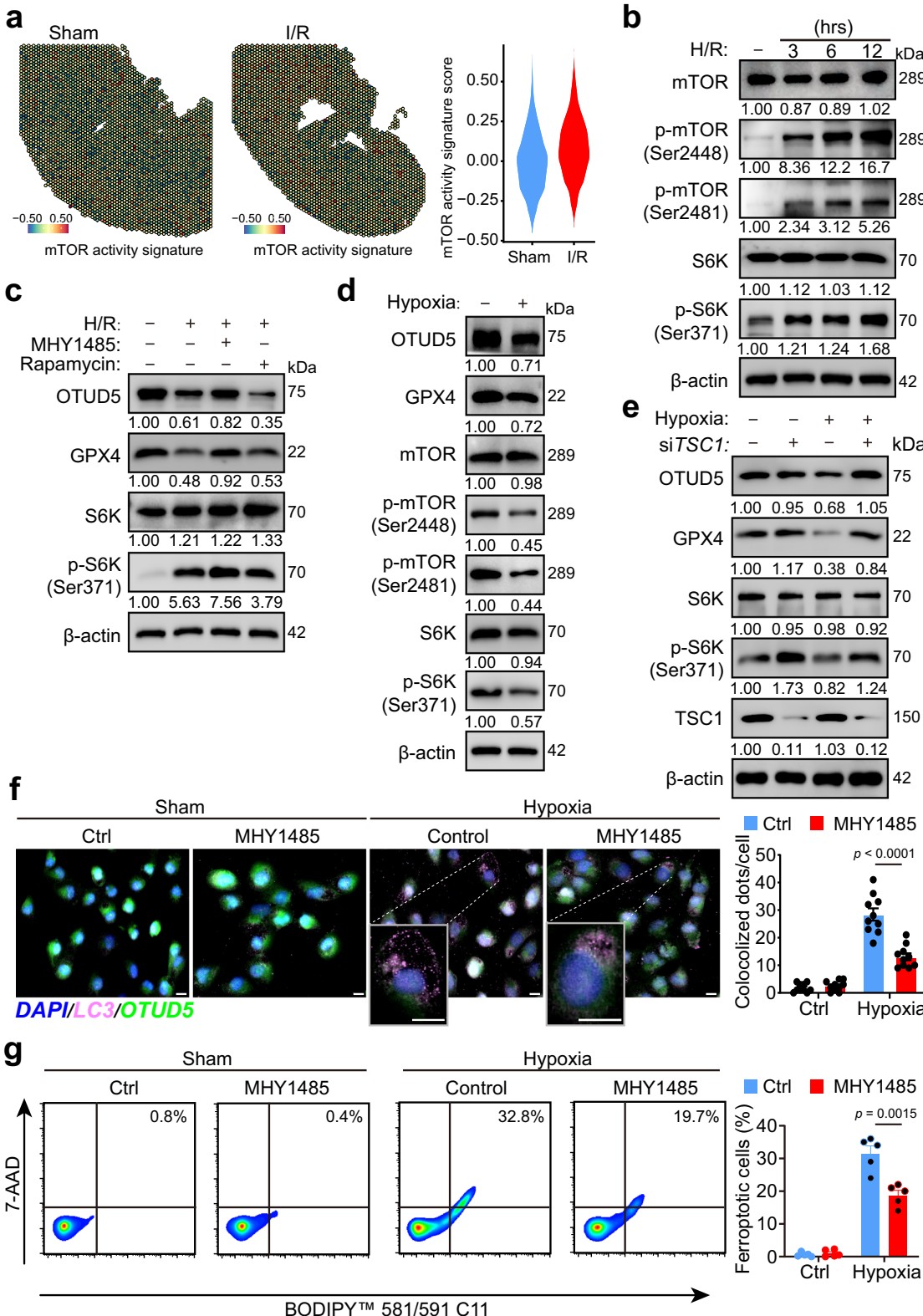

**Fig. 6 | H/R reduces OTUD5 through repressing mTORC1 signaling. a** Spatial feature plot and Violin plot of mTOR activity signature score in ST spots of the two groups. **b** PRTCs were treated by H/R for 3, 6, or 12 h and then collected and analyzed by immunoblotting. **c** Cells were treated by H/R in the presence of mTOR activator MHY1485 or mTOR inhibitor rapamycin for 3 h and then collected and analyzed by immunoblotting. **d** HK2 cells were treated with hypoxia condition for 6 h and then collected and analyzed by immunoblotting. **e** HK2 cells were transfected with siControl or si*TSC1* for 48 h and treated with or without hypoxia condition. Cells were collected and analyzed by immunoblotting. **f, g** HK2 cells were treated with hypoxia conditions in the presence or absence of mTOR activator MHY1485. And cells were collected for ferroptosis detection using flow cytometry. And cells were collected for detecting OTUD5's autophagy (**f**, $n = 10$ independent cells) using IF staining, and cell ferroptosis (**g**, $n = 5$ independent experiments) using flow cytometry. The magnification images show the degrading OTUD5 in the lysosome (LC3). Scale bar, 20 μm.; Data are presented as mean ± s.e.m., statistical significance between groups, as indicated, was determined using an unpaired two-tailed Student's *t*-test.

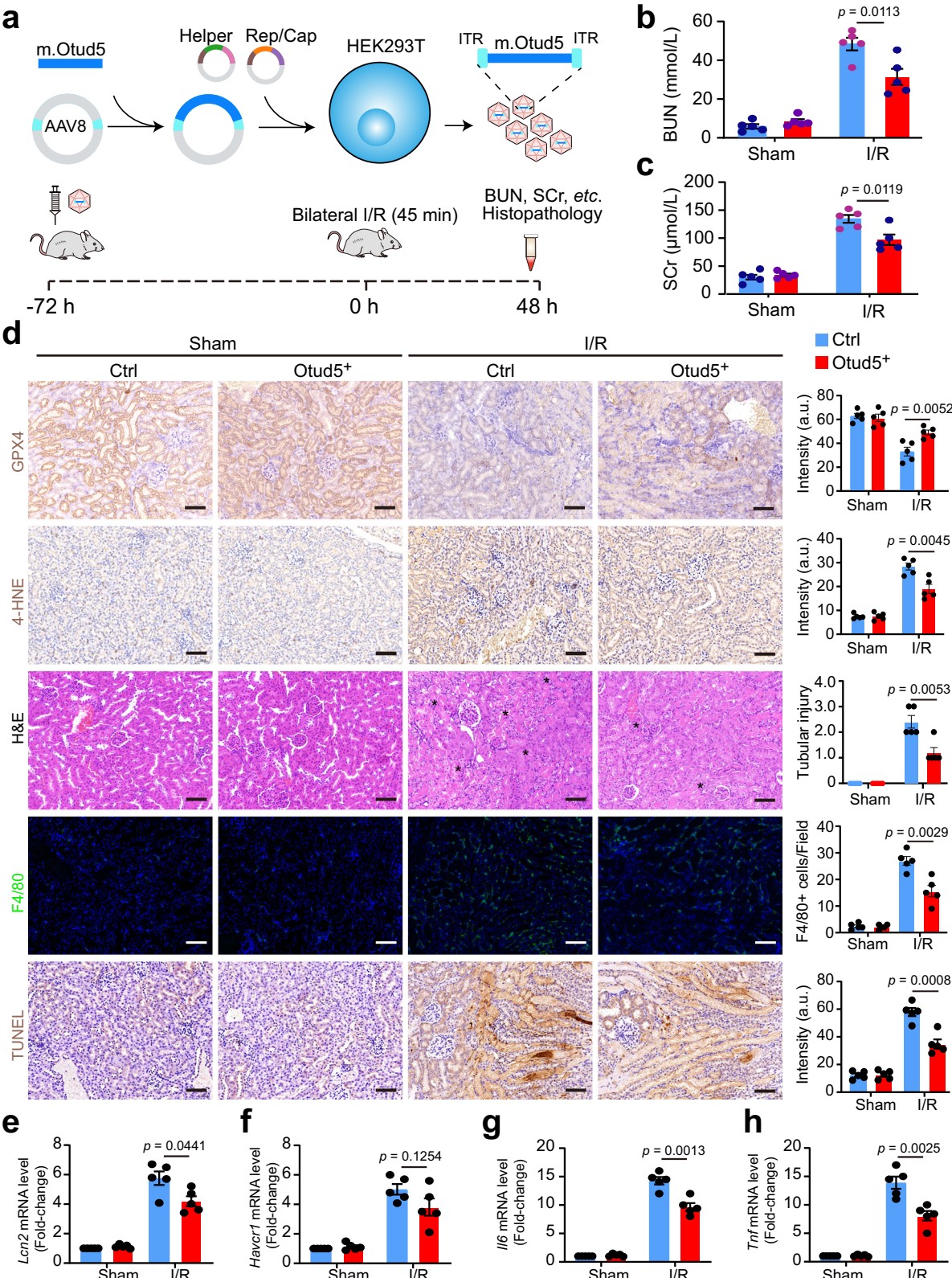

incubated for 2 h, and the absorbance at $OD_{450}$ was measured using a Multiskan™ FC microplate reader (Thermo Fisher Scientific).

## Histopathological analysis

The mouse kidney was embedded in paraffin and 4-μm tissue sections were prepared for IHC staining as previously described[46]. Briefly, tissue slides were deparaffinized in xylene and rehydrated in decreasing concentrations of ethanol. Sample slides were incubated with 3% $H_2O_2$ to deactivate endogenous peroxidase and heated at 95 °C for 30 min for antigen retrieval. Subsequently, the slides were incubated with 5% blocking serum for 15 min. The sections were then incubated with primary antibodies (see Supplementary Table 1) at 4 °C overnight, followed by incubation with a secondary antibody conjugated to horseradish peroxidase for 30 min. Tissue sections were

**Fig. 7 | AAV-mediated OTUD5 therapy protects renal function against I/R injury. a** Schematic shows the generation of AAV-*Otud5* virus. **b** 4–6-week-old C57BL/6J mice were intravenously injected with a single dose of $3 \times 10^{11}$ copies of *Otud5*-contained virus. 48 h after injection, mice were subjected to I/R surgery and lived for another 48 h. **b, c** Levels of BUN and SCr were analyzed in control and *Otud5*-expressed mice before and 48 h after I/R surgery (*n* = 5). **d** Kidneys were collected and subjected to H&E, IHC staining, immunofluorescence analysis, TUNEL staining, and quantification for evaluating kidney injury score, GPX4, and 4-HNE

expression, F4/80 expression, and cell death, respectively. Representative images of IHC staining for GPX4 and 4-HNE, H&E, F4/80-positive cells, and TUNEL staining in kidney sections from control and Otud5-expressed mice (*n* = 5); Scale bars, 50 μm. The asterisk indicates the injured tubular area. a.u.: arbitrary units. **e–h** The mRNA levels of *Lcn2, Havcr1, and Tnf* were analyzed in the kidneys of control and *Otud5*-expressed mice before and 48 h after I/R surgery. Data are mean ± s.e.m., *n* = 5; statistical significance between groups as indicated was determined using an unpaired two-tailed Student's *t*-test.

counterstained with hematoxylin and visualized with a microscope to obtain digital images (TCS SP5, Leica). For immunofluorescence, tissue or cell sections were fixed with 4% paraformaldehyde (Beyotime, Shanghai) at 37 °C for 10 min, and then permeabilized with 0.2% Triton X-100 (Beyotime, Shanghai) at 4 °C for 10 min. After blocking with 10% BSA and 2% normal goat serum in PBS for 1 h, the sections were incubated with the indicated primary antibodies for 2 h at room temperature or overnight at 4 °C. Following this, the sections were washed three times with 1 × PBS (Beyotime, Shanghai), and stained with fluorescence-conjugated secondary antibodies (Invitrogen) for 1 h at room temperature. Cell images were captured with a digital microscope (TCS SP5, Leica).

### Immunoblot and Immunoprecipitation

For immunoblot, kidney tissue or cells were homogenized using RIPA lysis buffer (R0010, Solarbio, Beijing, China) containing protease and phosphatase inhibitors cocktail (04693116001, Roche) and were sonicated. The protein concentrations were measured with BCA assay (Solarbio). Protein samples were separated using 10% SDS page gels and transferred onto a PVDF membrane (R1SB07718, Millipore). After blocking using 5% skimmed milk for 1 h at room temperature, the membrane was incubated with primary antibodies (Supplementary Table 1) at 4 °C overnight and then with the corresponding secondary antibodies for 1 h at room temperature. The membrane was washed three times with 1 × PBST between incubations. Proteins were visualized by enhanced chemiluminescence (ECL, 32209, Thermo Fisher Scientific). For Immunoprecipitation, cells were lysed on ice with NP40 lysis buffer (Solarbio) containing protease and phosphatase inhibitors cocktail (04693116001, Roche) for 30 minutes. After centrifugation at 12,000×*g* for 20 min at 4 °C, the supernatant was collected and incubated with protein A/G beads (Sant Cruz Biotechnology) for 1 h at 4 °C, followed by another round of centrifugation. The supernatant was collected and incubated with the indicated antibody (Supplemental Table 1) overnight at 4 °C with rotation. Protein A/G beads were then added to the supernatants and incubated for 4 h at 4 °C with rotation, followed by 5 times washing using NP40 lysis buffer. The beads were mixed with 2 × SDS loading buffer and boiled at 95 °C for 10 min. After centrifugation at 800×*g* for 5 min, the protein-contained supernatant was analyzed by immunoblotting as indicated above. The quantification of protein expression was analyzed by Image J (v1.47).

### Flow cytometry

For cell ferroptosis analysis, HK2 cells with or without hypoxia treatment were collected and washed twice with pre-chilled 1 × PBS. The cells were then resuspended in a staining buffer (420201, BioLegend) and stained with BODIPY 581/591 C11 at room temperature for 30 min. After washing and staining with 7-AAD, the cells were resuspended in a cell staining buffer and subjected to flow cytometry analysis. Fluorescent signals were acquired using a Beckman Coulter Gallios Flow Cytometer and analyzed using FlowJo software (v10.6).

### Detection of lipid peroxidation

For lipid peroxidation analysis, cells were stained with Liperfluo (L248, Dojindo, Kumamoto, Japan) in PBS containing 2% (w/v)

bovine serum albumin (BSA) on ice for 30 min according to the manufacturer's instructions. The fluorescent signals were acquired and visualized with a fluorescence microscope (Olympus). For total glutathione (GSH) and GSH/GSSG ratio detection, mouse kidneys were lysed using RIPA lysis buffer (R0010, Solarbio, Beijing, China) containing a protease and phosphatase inhibitor cocktail (04693116001, Roche). The content of total glutathione and oxidized glutathione (GSSG) was quantified in cell or tissue lysates according to the instructions of the GSH and GSSG Assay Kit (S0053, Beyotime, Shanghai, China).

### Quantitative real-time polymerase chain reaction (qRT-PCR)

Total RNA was extracted from homogenized kidneys or cells using Trizol Reagent (Invitrogen) and digested by RNase-free DNase I (Promega) following the manufacturer's instructions. A total of 2 μg RNA from each sample was reverse-transcribed into cDNA according to the High-Capacity cDNA Reverse Transcription Kit's instructions (4368814, Applied Biosystems™). Quantitative PCR was performed using the qPCR SYBR Green Master Mix (Takara) and analyzed using a LightCycler 96 real-time system (Roche Diagnostics). Relative quantification was calculated using the $2^{-\Delta\Delta Ct}$ method and normalized to GAPDH. The primer was commercially synthesised (Beijing Tsingke Biotech Co., Ltd.) and sequences are listed in Supplementary Table 2.

### Liquid chromatography–mass spectrometry (LC–MS/MS)

HK2 cells were collected in a lysis buffer containing a protease and phosphatase inhibitor cocktail (04693116001, Roche), incubated on ice for 30 min, and then centrifuged at 12,000×*g* for 15 min at 4 °C. The supernatant cell lysates were obtained and immunoprecipitated with an anti-GPX4 antibody overnight at 4 °C. The beads were washed three times with lysis buffer, boiled with SDS loading buffer, and subjected to SDS–PAGE. Gels with total proteins were stained with Coomassie Blue (ST030, Beyotime, Shanghai, China), excised, and digested with 10 ng/μl trypsin at 37 °C overnight. After enzymatic hydrolysis, the peptide mixtures were collected, desalted, and prepared for liquid chromatography-tandem mass spectrometry (LC–MS/MS) using a Q Exactive HFX platform (Thermo Scientific). The raw files generated from the spectrometer were searched against the UniProt Homo sapiens database (v2015-11-11) using Proteome Discoverer (Thermo Scientific, v.2.1).

### Gene transfection

For plasmid transfection, human WT or C224S OTUD5 plasmids were constructed by amplifying the corresponding cDNA and cloning it into pcDNA3.1(+)/myc-His A (V80020, Invitrogen) using the PCR method. Then, plasmids were transfected into HK2 cells with Lipofectamine 3000 (Invitrogen) according to the manufacturer's instructions. For siRNA transfection, cells were plated at a concentration of $5 \times 10^5$ cells per well in a six-well plate to achieve ~60% confluence. Cells were transfected with specific siRNA (ZIXI Biotech. Co., Ltd., Beijing, China) at a final concentration of 50 nmol/L with Lipofectamine 3000 (Invitrogen) in Opti-MEM (Invitrogen) according to the manufacturer's instructions. The cell medium was replaced with a complete medium after 6 hours of transfection, and cells were further cultured for 48 h.

### AAV preparation and gene delivery

AAV virus preparation and renal orthotopic injection were performed according to a previous study[47]. Briefly, the cDNA of mouse *Otud5* was synthesized, and AAV8-Otud5 was constructed using an AAV8 vector plasmid, an adenovirus helper plasmid, and an AAV helper plasmid, which were transiently transfected into 293T cells. The recombinant virus was harvested 72 h after transfection. The crude viral lysate was then purified by fractionation with iodixanol-gradient centrifugation. Viral titers were determined, and vectors that passed quality control were aliquoted and stored at −80 °C until use. For gene delivery, 4 to 6-week-old C57BL/6J mice were intravenously injected with AAV particles at $3 \times 10^{11}$ copies/ml per mouse. The injected mice were fed and monitored for three days after I/R induction.

### Library construction and single-cell mRNA sequencing

For single-cell suspension preparation, freshly collected mouse kidneys were rinsed twice with RPMI1640 (Gibco), minced into $3-5 \, mm^3$ pieces, and subjected to enzymatic digestion using Multi Tissue Dissociation Kits (130-110-204, Miltenyi Biotec) for 30 min under rotation at 250 rpm/min. Afterward, the cell suspension was filtered and centrifuged at $300 \times g$ at 4 °C. The cell pellet was collected, resuspended in 1 mL RBC lysis buffer (Sigma) to remove RBCs, and then washed twice with RPMI1640. Cell viability was assessed using Trypan Blue (Gibco), and dead cells were removed using magnetic bead separation with a Dead Cell Removal Kit (130-090-101, Miltenyi Biotec) if the cell viability was <85%. At least 20,000 cells per sample were applied to a single-cell master mix with lysis buffer and reverse transcription reagents, following the User Guide of 10X Genomics Chromium Single Cell 3' Reagent Kits (10X Genomics). The libraries were subjected to high-throughput sequencing on a NovaSeq6000 platform.

### Data processing and analysis of single-cell RNA-seq data

Raw sequencing data were processed with the Cellranger pipeline (version 3.1.0, 10X Genomics) and mapped to the murine reference genome to generate digital gene expression (DGE) matrices. The batch effects were corrected using the R package "harmony" (v1.2.0.1). Cells from different batches were merged, and expressions were normalized to generate a final gene expression matrix. A Seurat object was created, and the gene-barcode count matrices were analyzed using the R package "Seurat" (v4.3.0) following the pipeline (http://satijalab.org/seurat/). For quality control, cells with fewer than 200 genes or with mitochondrial gene percentages over 20% were filtered out from downstream analyses. For dimension reduction, the most variable genes were identified using the FindVariableGenes function (Seurat version 4.3.0), followed by a principal component analysis (PCA). The top 20 PCs were used for unsupervised clustering analysis and visualized using Uniform Manifold Approximation and Projection (UMAP) for dimension reduction. The FindAllMarkers function (Seurat version 4.3.0) was used to determine cluster-specific gene markers. Annotations were performed by mapping the marker genes of each cluster to canonical kidney cell type marker genes from previous studies[48,49]. For Gene Set Enrichment Analysis on PT cells, the gene expression matrix of PT from two groups was collected, and differentially expressed genes (DEGs) were analyzed using the FindMarkers function (Seurat version 4.3.0). GSEA analysis was performed using the R package "fgsea" (1.24.0) by mapping genes to the murine reference database org.Mm.eg.db (3.16.0).

### Spatial transcriptomics assay

Mouse kidneys were snap-frozen in liquid nitrogen and embedded in OCT (Tissue-Tek) for preparation of 10-µm cryosections. For tissue visualization, frozen tissue sections were fixed with pre-chilled methanol on the Visium Tissue Optimization Slides (10X Genomics, PN-1000193), followed by staining with hematoxylin (S3309, Dako)

and eosin (CS701, Dako). Brightfield histological images were taken with a Leica DMI8 whole-slide scanner. For RNA isolation and reverse transcription, sections were permeabilized with permeabilization enzymes to release mRNA, which was captured by probes on the Visium spatial gene expression slides (PN-1000184,10X Genomics). The captured mRNA was reversely transcribed to cDNA, spatially barcoded, amplified, and subjected to library construction using the Visium Spatial Library Construction Kit (PN-1000184,10X Genomics). Briefly, 10 µl of amplified cDNA from each sample was taken for library preparation through the processes of fragmentation, adapter ligation, PCR, and purification. The constructed library was sequenced with an Illumina Novaseq6000 sequencer with a sequencing depth of at least 100,000 reads per spot (performed by CapitalBio Technology, Beijing).

### Spatial transcriptomics analysis

The raw data from the spatial transcriptomics (ST) assay were processed using SpaceRanger (v2.0.1) to generate the ST spots matrix by performing a series of procedures, including alignment, filtering, barcode counting, and UMI counting. Spots with fewer than 200 genes were excluded from further analysis. Normalization across spots was performed with the SCTransform function (0.3.5), and anchors between the spatial transcriptomics spots were identified using the R package "Seurat" (v4.3.0) in R (v4.2.2). Dimension reduction and clustering were performed using PCA. The first ten PCs were used to generate clusters. For scoring on specific signaling, we performed the AddModuleScore function with default parameters in Seurat to determine the score by using the signaling signature genes, which are driver or marker genes of ferroptosis[50], or genes from the GSEA dataset (https://www.gsea-msigdb.org/gsea/msigdb/index.jsp), which are positively involved in autophagy regulation or mTOR signaling activation. For spatial gene/feature colocalization, the interested genes or features were inferred.

### Statistical analysis

Statistical analyses were performed using Prism 8 (GraphPad Software, San Diego, CA, USA). *p* values were calculated by a two-tailed unpaired Student's *t*-test between two groups. *p*-value <0.05 was considered significant.

### Reporting summary

Further information on research design is available in the Nature Portfolio Reporting Summary linked to this article.

## Data availability

The single-cell RNA-Seq and spatial transcriptomic data generated in this study have been deposited in the National Center for Biotechnology Information Gene Expression Omnibus database with accession code GSE244330. Uncropped blots and other source data are available in the Source Data file. Further information and requests for resources should be directed to the corresponding authors. Source data are provided with this paper.

## Code availability

R scripts used in this study are available on GitHub (https://github.com/Toby111/scRNA-spatial-code)

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

## Acknowledgements

We thank Min-Xuan Sun (Suzhou Institute of Biomedical Engineering and Technology, Chinese Academy of Sciences) and all members of his laboratory for insightful comments and assistance on paper preparation.

Funding: This work was supported by the National Natural Science Foundation of China (82270713 to J.L.); Key Laboratory Foundation of Structural Deformities in Children of Suzhou (SZS2022018 to X.M.Y.); Gusu Health Talent Project of Suzhou (GSWS2022058 to J.L., GSWS2020049 to X.M.Y.). Suzhou Science and Technology Development Plan Project (SKY2023059 to J.L., SKY2023002 to X.M.Y., SKY2023194 to Q.W.X.).

## Author contributions

L.K.C., X.C., L.W., and Q.D. performed a large part of cell and immunoblot experiments, contributed to overall experiments, and analyzed the data. X.Q.D. and Y.K. performed immunostaining and FACS experiments. Y.M.M. helped with data analysis of single-cell RNA sequencing and spatial transcriptomics. Y.Z., L.L.Y., Q.W.X., M.C.F., and T.Z. performed siRNA and plasmid experiments. H.T.Z., S.Z.C., and Z.R.M. assisted with Pax8Cre and Otud5fl/fl mice breeding and animal experiments. S.W.H. and R.W. provided essential materials and helped with manuscript preparation. J.L. C.H.C. and X.M.Y. conceived the project. J.L. and X.M.Y. directed the experiment. J.L. and C.H.C. wrote the manuscript. All the authors commented on the manuscript draft.

## Competing interests

The authors declare no competing interests.
