## [Peer Review File · Nature Communications]

Autophagy of OTUD5 Destabilizes GPX4 to Confer Ferroptosis-Dependent Kidney InjuryREVIEWER COMMENTS

Reviewer #1 (Remarks to the Author):

This study explored the spatial distribution of ferroptosis mediated by GPX4 based on spatial transcriptomics, found the mechanism of post-transcriptional degradation of GPX4, and further identified the regulatory role of OTUD5 on GPX4 ubiquitination. Subsequent single-cell transcriptome and Bulk transcriptome sequencing showed that autophagy related pathways were significantly enriched after H/R, revealing the existence of endocytogenic autophagy degradation of OTUD5, and confirming that mTORC1 activity regulated the H/ R-induced autophagy degradation of OTUD5. The regulatory pathway identified by the author had not been reported in previous studies. The design of this study is reasonable, the experimental method is advanced, the research content is comprehensive, and the obtained results are real and reliable. The regulatory pathway discovered by the author is innovative and reported in the past, which provides a new direction for the treatment of renal ischemia-reperfusion injury.

I have some comments as followed.

1. There are problems with the marking of two pictures in the article. Please check whether there are errors in the significance marking in Figure S3a and the digital marking of the protein imprint in Figure 7B.
2. The text in the heat map in Figure S2b is too small, so it is suggested that the author enlarge the font or show the differential expression multiples and p-values of UPS/DUBs-associated genes in a tabular form.
3. The authors indicate in the second paragraph of result 2 (line 153-168, page 2) that ubiquitin-dependent protein degradation regulation of GPX4 must be due to changes in UPS/DUBs content, which is inaccurate because UPS/DUBs also has many post-translational modifications, such as phosphorylation and fatty acylation. This also affects the change of function without changing the UPS/DUBs content.

Reviewer #2 (Remarks to the Author):

In the manuscript Autophagy of OTUD5 Destabilizes GPX4 to Confer Ferroptosis-Dependent Kidney Injury, Chu et al. describe the mechanism of GPX-4 mediated ferroptosis via

degradation of the deubiquitinating protein OTUD5, potentially through mTORC1-mediated autophagy. Overall is a well designed and written manuscript, and the authors build a strong case in favor of targeting the OTUD5 / GPX4 pathway to facilitate recovery in AKI. The hypothesis for the implication of the ferroptosis pathway and GPX4 is derived from single cell RNA sequencing and spatial transcriptomics. I describe my concerns with this methodology below:

- The single cell seems to present a strong batch effect regarding cell types annotation. This data is the basis for the ferroptosis pathway involvement. For instance, PT, CD-PC and CD-IC do not seem to cluster together between the Sham and I/R samples. Combined with the complete absence of LOH in I/R and a cell population with around 30% of macrophages raise concerns regarding the cell dissociation of the I/R sample. The histological images presented through the manuscript seem to indicate preservation of epithelial structure in the I/R model. If immune cell infiltration is present to this degree, histological evidence could be presented. The authors should consider presenting the cell type marker genes (supplementary Fig 1c) broken down by specimen, and maybe by the original clusters as well. An immediate solution would be the inclusion of extra samples to validate the cell type change, but I don't think this would be necessary here, provided the cell type annotation questions are addressed in another fashion.
- It is not clear the role of spatial transcriptomics in this manuscript, besides localizing the expression of GPX4 and a few selected pathways. The authors identify up to 7 clusters in each sample, but no effort at annotating them is taken. Maybe using any reference based cell annotation could help validate the cell types identified in panels A and B. An example is Seurat's Transfer Anchors methodology. The authors describe the use of this algorithm in the methods section, but no results are presented. The application of this tool should be clarified. A potential use of this technology would be to investigate co-localization of expression of GPX4 and OTUD5, as well as other UPS/DUB proteins
- Could the authors elaborate on the selection of five UPS/DUB proteins from the 267 immunoprecipitated ones?
- Overall, could the authors comment on the advantage of using single-cell RNA sequencing and spatial transcriptomics in this manuscript. I believe similar insights to the ones presented here could be achieved with bulk RNA sequencing and immunofluorescence, potentially with more confidence in the results.

Reviewer #3 (Remarks to the Author):

The present study explores the downregulation of GPX4 expression in IRI kidneys and its contribution to ferroptosis vulnerability. The authors demonstrated that GPX4 reduction occurs via ubiquitin-proteasomal degradation in response to I/R injury, and they identified OTUD5 as a GPX4-interacting protein that promotes ferroptosis resistance during I/R by stabilizing GPX4 expression. This study addresses an interesting and important topic in the fields of nephrology and ferroptosis, and the study design is well controlled. However, there are several points that require clarification and additional experiments to support the authors' conclusions:

1. Of note, the authors use 4HNE as a marker of ferroptosis throughout the paper, but it is an indirect marker of lipid peroxidation and not a direct marker of ferroptosis. Please revise the text to accurately reflect this.

1-1. Since many previous studies have reported the involvement of ferroptosis in kidney IRI, only phrase change would be fine regarding the explanation of figure 1E.

1-2. In in vitro study, to demonstrate ferroptosis, presenting the following evidences are required: i) cell death, ii) lipid peroxidation, and iii) almost complete rescuing effect against the cell death by specific ferroptosis inhibitor. Thus, the evidence of ii) and iii) is missing for the experiment of the hypoxia/reperfusion condition. Please provide this evidence to support the claim of ferroptosis in the in vitro model.

1-3. In Figures 3 and 7, please include direct evidence of tubular cell death (e.g., TUNEL staining) and its quantification to demonstrate the effect of Otud5.

2. The effect of OTUD5 should be examined under general ferroptosis condition more in detail. In fig 4F, the effect of OTUD5 was not fully examined since only single dose of RSL3 was used. Multiple doses of RSL3 and other ferroptosis inducer(s) not targeting GPX4 (such as erastin) should be used to fully examine the effect of OTUD5, including both siOTUD5 and OTUD5 overexpression.

3. Please consider to include the data of 4HNE staining in figure 7E, as it is missing, and only the 4HNE WB data is shown.

4. The authors should explain why GPX4 expression at the basal level is different between siCtr and siOTUD5 in Fig 2J although the basal level was comparable in other figures (such as Fig1G).

5. To validate the Pax8cre-Otud5 model, the depletion of Otud5 protein expression level in the mice could be shown also by WB. Additionally, please present the GPX4 protein expression level by WB after IRI in the KO mice and control. The reviewer could not find the data.

6. For Figure 1H, clarify how the minor decrease in Gpx4 mRNA was quantified in the pseudo-color images, as it seems to show a significant decrease. In contrast, Figure S1H reports no difference in Gpx4 mRNA between sham and I/R. Explain the discrepancy in these results.

7. In Figures 7 and S6, please address the presence of the band detected by FLAG(Otub5) antibody in WT control mice. If these are nonspecific bands, annotation should be provided.

8. Figure 7B shows all relative values as 1.00, which is likely incorrect. Please review and correct this data.

Review Comments:

Reviewer #1 (Remarks to the Author):

This study explored the spatial distribution of ferroptosis mediated by GPX4 based on spatial transcriptomics, found the mechanism of post-transcriptional degradation of GPX4, and further identified the regulatory role of OTUD5 on GPX4 ubiquitination. Subsequent single-cell transcriptome and Bulk transcriptome sequencing showed that autophagy-related pathways were significantly enriched after H/R, revealing the existence of endocytogenic autophagy degradation of OTUD5, and confirming that mTORC1 activity regulated the H/ R-induced autophagy degradation of OTUD5. The regulatory pathway identified by the author had not been reported in previous studies. The design of this study is reasonable, the experimental method is advanced, the research content is comprehensive, and the obtained results are real and reliable. The regulatory pathway discovered by the author is innovative and reported in the past, which provides a new direction for the treatment of renal ischemia-reperfusion injury.

I have some comments as followed.

- 1. There are problems with the marking of two pictures in the article. Please check whether there are errors in the significance marking in Figure S3a and the digital marking of the protein imprint in Figure 7B.*

R: We thank the reviewer for pointing out the labeling issues in Figure S3c (originally Figure S3a in our initial submission) and Supplementary Fig. 7c (originally Figure 7B in our initial submission). We apologize for any confusion caused by these errors. To address the reviewer's concerns, we have meticulously reviewed and corrected the mistakes in the labeling of the figures mentioned. The revised versions of **Supplementary Fig. 3c** and **Supplementary Fig. 7c** are also incorporated below in this response letter. We hope that these revisions adequately address the concerns raised and enhance the clarity and accuracy of our manuscript.

Supplementary Fig. 3c. The mRNA level of *Gpx4* in *Pax8^{Cre}Otud5^{fl/fl}* mice and their WT littermates. All values are presented as mean±s.e.m.; *p* values were calculated by unpaired two-tailed student's *t*-test, **p*< 0.05.

Supplementary Fig. 7c. 4 to 6-week-old C57BL/6J mice were intravenously injected with a single dose of 3×10^{11} copies of *Otud5*-contained virus. 48 hours after injection, mice were subjected to I/R surgery and lived for another 48 hours. After that, mice were sacrificed and the kidneys were collected for immunoblot ($n=3$). Data are presented as mean \pm s.e.m.; Statistical significance was determined using an unpaired two-tailed Student's *t*-test, n.s.: no significance.

2. The text in the heat map in Figure S2b is too small, so it is suggested that the author enlarge the font or show the differential expression multiples and *p*-values of UPS/DUBs-associated genes in a tabular form.

R: We appreciate the helpful suggestions regarding Figure S2b. Acknowledging the difficulty in reading the text in the original heat map due to the small font size, we have taken measures to improve the clarity and accessibility of the information presented. In response to the suggestion, we have revised Figure S2b to incorporate a volcano plot. This format visually represents the data in a manner that facilitates easier interpretation of the differential expression of UPS/DUBs-associated genes. In the revised **Supplementary Fig. 2b** (also shown below), the most downregulated UPS/DUBs genes are now distinctly labeled in blue, while the most upregulated UPS/DUBs genes are labeled in red font. We believe this revised representation not only enhances readability but also provides a clearer and more immediate understanding of the gene expression data.

Supplementary Fig. 2b. The volcano plot shows the expression profile of UPS/DUBs-associated genes in the kidney before and after I/R induction (GSE87024), the most dysregulated UPS/DUBs genes were labeled in blue (downregulation) or red font (upregulation).

3. *The authors indicate in the second paragraph of result 2 (line 153-168, page 2) that ubiquitin-dependent protein degradation regulation of GPX4 must be due to changes in UPS/DUBs content, which is inaccurate because UPS/DUBs also has many post-translational modifications, such as phosphorylation and fatty acylation. This also affects the change of function without changing the UPS/DUBs content.*

R: We deeply appreciate the reviewer's constructive comment and agree with the observation that post-translational modifications, including phosphorylation and fatty acylation, can influence the functionality of UPS/DUBs without altering their content. We recognize the importance of this nuanced distinction for accurately interpreting and presenting our findings. To provide clarity, our study examines changes in the expression of downstream target proteins resulting either from alterations in UPS/DUBs content or from the impact of post-translational modifications on UPS/DUBs functionality. To prevent any misunderstanding or oversimplification, we have carefully reviewed and revised the relevant sections of our manuscript. Specifically, we removed the inaccurate statement at the beginning of paragraph 2 in the Results section, emphasizing instead that our findings primarily concern the expression of downstream target proteins, with consideration given to the potential impact of post-translational modifications.

Reviewer #2 (Remarks to the Author):

In the manuscript Autophagy of OTUD5 Destabilizes GPX4 to Confer Ferroptosis-Dependent Kidney Injury, Chu et al. describe the mechanism of GPX-4 mediated ferroptosis via degradation of the deubiquitinating protein OTUD5, potentially through mTORC1-mediated autophagy. Overall is a well-designed and written manuscript, and the authors build a strong case in favor of targeting the OTUD5/GPX4 pathway to facilitate recovery in AKI. The hypothesis for the implication of the ferroptosis pathway and GPX4 is derived from single-cell RNA sequencing and spatial transcriptomics. I describe my concerns with this methodology below:

Comment 1: The single cell seems to present a strong batch effect regarding cell types annotation. This data is the basis for the ferroptosis pathway involvement. For instance, PT, CD-PC, and CD-IC do not seem to cluster together between the Sham and I/R samples. Combined with the complete absence of LOH in I/R and a cell population with around 30% of macrophages raise concerns regarding the cell dissociation of the I/R sample. The histological images presented through the manuscript seem to indicate preservation of epithelial structure in the I/R model. If immune cell infiltration is present to this degree, histological evidence could be presented. The authors should consider presenting the cell type marker genes (supplementary Fig 1c) broken down by specimen, and maybe by the original clusters as well. An immediate solution would be the inclusion of extra samples to validate the cell type change, but I don't think this would be necessary here, provided the cell type annotation questions are addressed in another fashion.

R: We are grateful to the reviewer for the insightful comments and suggestions. In light of this feedback, we revisited our single-cell data and implemented corrective measures. Specifically, we addressed batch effects prior to cell clustering using the "harmony" package in R, as depicted in **Supplementary Fig. 1c** (also shown below). Following the suggestion, we have further detailed the cell type marker genes, presenting them broken down by specimen in **Supplementary Fig. 1d** (please see the figure below).

Upon re-evaluation, we observed a significant increase in the percentage of macrophages within the I/R samples, accounting for 22.8%, in comparison to a mere 2.0% in the sham samples (as depicted in Fig. 1b). Admittedly, this proportion is unexpectedly high, deviating from conventional understanding. To corroborate our findings, we analyzed the public datasets GSE139506 and GSE197626 and identified similar trends: the former exhibited an increase in macrophage percentage from 1.6% in sham samples to 10.8% in I/R samples (PMID: 33115917), and the latter revealed a substantial increase from 4.3% in sham samples to 33.2% in I/R samples (PMID: 35986026).

Despite the single-cell analysis findings, the extent of immune cell infiltration was not mirrored in our histological images. We hypothesize that this discrepancy may stem from the suspension preparation phase. In endeavoring to achieve the necessary cell numbers and viability for sequencing, dead cells, predominantly renal tubular

cells, which constitute a large portion of kidney cells, may have been inadvertently excluded. This exclusion could result in an overrepresentation of immune cells in our single-cell sequencing data compared to the actual tissue composition. We hope that our revisions and explanations satisfactorily address the concerns.

Supplementary Fig. 1c. The batch effect correction of the two samples.

Supplementary Fig. 1d. The dot plot shows expressions of two representative marker genes of each cell type.

Comment 2: It is not clear the role of spatial transcriptomics in this manuscript, besides localizing the expression of GPX4 and a few selected pathways. The authors identify up to 7 clusters in each sample, but no effort at annotating them is taken. Maybe using any reference-based cell annotation could help validate the cell types identified in panels A and B. An example is Seurat's Transfer Anchors methodology. The authors describe the use of this algorithm in the methods section, but no results are presented. The application of this tool should be clarified. A potential use of this technology would be to investigate the co-localization of expression of GPX4 and OTUD5, as well as other UPS/DUB proteins

R: We appreciate the reviewer's detailed feedback regarding the role of spatial

transcriptomics in our manuscript. In line with the reviewer’s suggestion, we have indeed made efforts to provide spatial annotations and display the identified cell types in spatial spots using the Seurat’s Transfer Anchors methodology, which is detailed in the figure provided below. In our study, we utilized this methodology to analyze the ferroptosis signature (as shown in **Fig. 1d**), as well as to investigate the co-localization of the autophagy signature and *Otud5* expression (depicted in **Fig. 5c and d**). Additionally, as the reviewer pointed out, the co-localization of *Gpx4* and *Otud5* expression has been analyzed and presented in **Supplementary Fig. 2d**. We hope our revisions provide a clearer understanding of our approach and findings.

Spatial cluster annotation, localization, and the signature score of each cell type in the spatial transcriptomics spots.

Fig. 1d. Spatial feature plots and violin plots of the ferroptosis signature score in ST spots.

Supplementary Fig. 2d. Feature plot of *Otud5* and *Gpx4* expression, and their colocalization in ST spots.

Comment 3: Could the authors elaborate on the selection of five UPS/DUB proteins from the 267 immunoprecipitated ones?

R: We appreciate the reviewer's comment. From the pool of 267 immunoprecipitated proteins listed in Supporting Information Table 1 of the source data file, each protein was meticulously reviewed manually. Our review process involved confirming whether each protein was either a member of the UPS/DUB family or closely associated with ubiquitination and deubiquitination processes. Through this rigorous selection process, five proteins, namely OTUB1, OTUD5, TRIM21, UBR5, and XIAP, were identified and defined as members of the UPS/DUB family. These proteins were selected based on their known roles and associations within the ubiquitin-proteasome system (UPS) and deubiquitinating (DUB) processes, making them particularly relevant for our study's focus and analysis. We hope this clarification adequately addresses the question raised.

Comment 4: Overall, could the authors comment on the advantage of using single-cell RNA sequencing and spatial transcriptomics in this manuscript. I believe similar insights to the ones presented here could be achieved with bulk RNA sequencing and immunofluorescence, potentially with more confidence in the results.

R: We thank the reviewer for the constructive comment. We concur that bulk RNA sequencing and immunofluorescence are valuable techniques; however, we opted for single-cell RNA sequencing and spatial transcriptomics for specific advantages. Single-cell RNA sequencing offers a finer resolution by distinguishing between different cell types within the kidney. This allows us to accurately characterize and analyze the transcriptome of specific cell types of interest, such as proximal tubules (PT) in this study. For instance, it provided crucial insights into how PT cells may undergo ferroptosis in response to ischemia/reperfusion (I/R), while also paving the

way for future explorations into changes in gene expression across various cell types.

Spatial transcriptomics, on the other hand, not only delivers precise transcriptome data but also aids in localizing specific cells or genes within the kidney tissue, like GPX4 in our case. This approach revealed the spatial distribution of GPX4 and suggested its reduction in response to I/R, likely regulated through post-transcriptional modifications rather than transcriptional changes.

While bulk RNA sequencing paired with immunofluorescence could yield similar insights, the combination of single-cell RNA sequencing and spatial transcriptomics offers richer and more accurate data. The depth and precision afforded by these advanced techniques provide a more comprehensive understanding, which we believe is pivotal for the integrity and robustness of our study's findings. Hence, we consider these methodologies not only appropriate but advantageous for our research objectives.

Reviewer #3 (Remarks to the Author):

The present study explores the downregulation of GPX4 expression in IRI kidneys and its contribution to ferroptosis vulnerability. The authors demonstrated that GPX4 reduction occurs via ubiquitin-proteasomal degradation in response to I/R injury, and they identified OTUD5 as a GPX4-interacting protein that promotes ferroptosis resistance during I/R by stabilizing GPX4 expression. This study addresses an interesting and important topic in the fields of nephrology and ferroptosis, and the study design is well-controlled. However, there are several points that require clarification and additional experiments to support the authors' conclusions:

- 1. Of note, the authors use 4HNE as a marker of ferroptosis throughout the paper, but it is an indirect marker of lipid peroxidation and not a direct marker of ferroptosis. Please revise the text to accurately reflect this.*

R: We appreciate the detailed feedback provided by the reviewer and have taken the necessary steps to address each point raised. We acknowledge the reviewer's perspective on 4-hydroxynonenal (4-HNE) and have made revisions throughout our manuscript to accurately reflect its role as a marker for lipid peroxidation, rather than a direct marker for ferroptosis.

- 1-1. Since many previous studies have reported the involvement of ferroptosis in kidney IRI, only a phrase change would be fine regarding the explanation of Figure 1E.*

R: We thank the reviewer for highlighting this. We have now modified the description by removing the phrase "4-hydroxynonenal (4-HNE), a marker for ferroptosis". This amendment can be found on line 113 of page 3 of our revised manuscript.

- 1-2. In in vitro study, to demonstrate ferroptosis, presenting the following evidences are required: i) cell death, ii) lipid peroxidation, and iii) almost complete rescuing effect against the cell death by specific ferroptosis inhibitor. Thus, the evidence of ii) and iii) is missing for the experiment of the hypoxia/reperfusion condition. Please provide this evidence to support the claim of ferroptosis in the in vitro model.*

R: We concur with the reviewer's emphasis on the importance of presenting comprehensive evidence for ferroptosis. To bolster our claims, we conducted experiments using the ferroptosis inhibitor, Ferrostatin-1 (Fer-1). With the phospholipid peroxidation dye Liperfluo, it was observed that H/R-induced ferroptosis could be counteracted by Fer-1 (as seen in **Fig. 1i**). Additionally, flow cytometry analyses with lipid peroxidation sensor BODIPY™ 581/591 C11 and cell death Dye 7-AAD confirmed that Fer-1 reversed H/R-induced ferroptosis, as demonstrated in **Fig. 1j**. These results solidify our findings that H/R induces renal tubular cell ferroptosis.

Fig. 1i and j. **i** Representative images and quantification of cell membrane lipid peroxidation stained by the liperfluor probe; scale bars, 50 μ m. **j** Ferroptosis was measured using fluorescent lipid peroxidation sensor BODIPY™ 581/591 C11 and cell death Dye 7AAD in HK2 cells treatment with or without Fer-1 after H/R induction. Error bars represent mean \pm s.e.m., $n = 3$; statistical significance was determined using an unpaired two-tailed Student's t-test. * $p < 0.05$.

1-3. In Figures 3 and 7, please include direct evidence of tubular cell death (e.g., TUNEL staining) and its quantification to demonstrate the effect of *Otud5*.

R: In response to the request for direct evidence of tubular cell death, we performed TUNEL staining. The results and quantification can now be viewed in **Fig. 3b** and **Fig. 7d** (also shown below).

Fig. 3b 4 to 6-week-old *Pax8^{Cre} Otud5^{fl/fl}* mice and their WT littermates ($n = 5$) were subjected to kidney I/R surgery. After 48 hours, kidneys were collected and subjected to TUNEL staining and quantification for cell death; Scale bars, 50 μ m.

Fig. 7d Kidneys were collected and subjected to TUNEL staining and quantification for cell death. Representative images of IHC staining for TUNEL staining in the kidney section from control and *Otud5*-expressed mice; Scale bars, 50 μ m.

2. The effect of OTUD5 should be examined under general ferroptosis conditions more in detail. In Fig 4F, the effect of OTUD5 was not fully examined since only a single dose of RSL3 was used. Multiple doses of RSL3 and other ferroptosis inducer(s) not targeting GPX4 (such as erastin) should be used to fully examine the effect of OTUD5, including both siOTUD5 and OTUD5 overexpression.

R: We thank the reviewer for the insightful comment. To thoroughly examine the effect of OTUD5 under various ferroptosis conditions, we conducted additional experiments using both siOTUD5 and OTUD5 overexpression in renal tubular cells. We exposed these cells to different doses of RSL3 as well as another ferroptosis inducer, erastin, that does not target GPX4. The results of these experiments are presented in Fig. 4f and Supplementary Fig. 4a-c, as shown below. These additional data offer a more comprehensive understanding of OTUD5's role under various ferroptotic conditions.

Fig. 4f Cell ferroptosis was measured in WT OTUD5-transfected cells in the presence or absence of different doses of GPX4 inhibitor RSL3 under H/R induction.

Supplementary Fig. 4. OTUD5 protects renal tubular cells from ferroptosis in response to H/R injury. **a** Cell ferroptosis was measured in EV or WT OTUD5-transfected cells in the presence or absence of Erastin with the indicated doses for 24 hours. **b** Cell ferroptosis was measured in WT OTUD5-transfected cells in the presence or absence of Erastin with the indicated doses for 24 hours under H/R induction. **c** Cell ferroptosis was measured in siOTUD5-transfected cells in the presence or absence of RSL3 with the indicated doses under H/R induction. Data are from three repeated experiments, and presented as mean ± s.e.m.; statistical significance between groups as indicated was determined using an unpaired two-tailed Student's *t*-test, *n.s.*: no significance, **p* < 0.05.

3. Please consider to include the data of 4HNE staining in figure 7E, as it is missing, and only the 4HNE WB data is shown.

R: We thank the reviewer for the helpful suggestion. In response, we have added the data showing 4-HNE expressions to **Fig. 7d** in the revised manuscript. Additionally, the Western Blot data for 4-HNE, previously located in Fig. 7e, has been moved to **Supplementary Fig. 7c** for reference. We appreciate the reviewer's attention to detail, and hope this addition clarifies the presentation of our data.

Fig. 7d Kidneys were collected and subjected to IHC staining for 4-HNE expression. Representative images of 4-HNE in the kidney section from control and *Otud5*-expressed mice ($n = 5$) with or without I/R induction; Scale bars, 50 μm .

4. The authors should explain why GPX4 expression at the basal level is different between *siCtrl* and *siOTUD5* in Fig 2J although the basal level was comparable in other figures (such as Fig1G).

R: We thank the reviewer for raising this important point. The discrepancy in GPX4 expression at basal levels between Fig. 2j and other figures, such as Fig. 1g, arises from the different conditions under which the experiments were conducted. Specifically, while Fig. 1g shows GPX4 expression under sham and I/R conditions, Fig. 2j presents GPX4 expression in cells transfected with either *siOtud5* or *siCtrl* under hypoxia/reoxygenation (H/R) conditions. Given these distinct experimental setups, it is plausible to observe a lower level of GPX4 in *siOtud5*-transfected cells compared to *siCtrl*-transfected cells under H/R conditions. We have clarified this in the figure legend to avoid any confusion.

5. To validate the *Pax8cre-Otud5* model, the depletion of *Otud5* protein expression level in the mice could be shown also by WB. Additionally, please present the GPX4 protein expression level by WB after IRI in the KO mice and control. The reviewer could not find the data.

R: We appreciate the insightful comments. To further validate the *Pax8cre-Otud5* model, we have incorporated Western Blot analysis data that show the expression levels of OTUD5 in the kidneys of both wild-type and *Pax8cre/Otud5^{fl/fl}* mice. This additional data provides a clearer understanding of OTUD5 depletion in the model. Furthermore, as per the request, we have also included the expression levels of GPX4 protein post-ischemia-reperfusion injury in knockout mice and their

corresponding controls. These data are presented in **Supplementary Fig. 3b** (also shown below for your reference). We hope that this additional information adequately addresses the concern raised.

Supplementary Fig. 3b 4 to 6-week-old *Pax8^{Cre}Otud5^{fl/fl}* and their WT littermates ($n = 4$) were subjected to kidney I/R surgery. After 48 hours, kidneys were collected and subjected to immunoblot analysis for OTUD5 and GPX4.

6. For Figure 1H, clarify how the minor decrease in *Gpx4* mRNA was quantified in the pseudo-color images, as it seems to show a significant decrease. In contrast, Figure S1H reports no difference in *Gpx4* mRNA between sham and I/R. Explain the discrepancy in these results.

R: We thank the reviewer for pointing out the need for clarification on this. While we acknowledge that the pseudo-color images in Fig. 1h might visually suggest a more significant decrease in *Gpx4* mRNA expression, our quantitative analysis of the spatial expression of *Gpx4* mRNA reveals only a slight reduction in the I/R group compared to the sham group (please refer to the right panel of Fig. 1h). This quantified data aligns with the results obtained from our qRT-PCR analysis presented in Supplementary Fig. 1j. The visual discrepancy may arise from the nature of pseudo-color images where variations in color intensities might not linearly correlate with the magnitude of change in expression levels, thus creating a visual impression of a significant change when the actual quantitative difference is minor. We hope this clarification resolves the perceived discrepancy between the figures.

Fig. 1h Spatial feature plots and violin plots of *Gpx4* in ST spots from sham or I/R-treated mouse kidneys.

7. In Figures 7 and S6, please address the presence of the band detected by FLAG(*Otub5*) antibody in WT control mice. If these are nonspecific bands, annotation should be provided.

R: We thank the reviewer for bringing this to our attention. We have reviewed the bands detected by the FLAG (*Otub5*) antibody in WT control mice in Figures 7 and S6. After careful assessment, we concur that these bands are likely nonspecific. We have provided annotations in **Supplementary Fig. 7a** and **7c** (also shown below) to clarify this point and prevent any misunderstanding or misinterpretation of the data. These annotations indicate that the bands in question are nonspecific, ensuring clarity and accuracy in the presentation of our results.

Supplementary Fig. 7. **a** 4 to 6-week-old C57BL/6J mice were intravenously injected with a single dose of 3×10^{11} copies of *Otud5*-contained virus. 48 hours after injection, mice were sacrificed and the kidneys were collected for immunoblot. The levels of GPX4 and Flag (OTUD5) were analyzed. **b** The mRNA level of *Gpx4* in the *Otud5*-expressed mice and the control mice ($n=3$). **c** 4 to 6-week-old C57BL/6J mice were intravenously injected with a single dose of 3×10^{11} copies of *Otud5*-contained virus. 48 hours after injection, mice were subjected to I/R surgery and lived for another 48 hours. After that, mice were sacrificed and the kidneys were collected for immunoblot ($n=3$). Data are presented as mean \pm s.e.m.; Statistical significance was determined using an unpaired two-tailed Student's *t*-test, *n.s.*: no significance.

8. *Figure 7B shows all relative values as 1.00, which is likely incorrect. Please review and correct this data.*

R: We appreciate the reviewer's attention to detail. Upon re-examination, we realized that the values in Figure 7B were inaccurately represented, and apologize for the oversight. We have now revised the figure to present the accurate quantification data. The corrected figure and its quantification information can be found in Supplementary Fig. 7c, as mentioned earlier. We believe this amendment addresses the issue appropriately.

REVIEWER COMMENTS

Reviewer #1 (Remarks to the Author):

In response to the comments made by the previous reviewer, the authors completed a large number of supplementary experiments, revised the manuscript in detail, and answered and clarified the questions raised by the reviewer. The authors used single-cell sequencing and spatial transcriptome methods, which showed the unique advantages of this experimental method although the cost was high, and successfully demonstrated the mechanism of GPX-4 mediating ferroptosis by degrading the deubiquitylated protein OTUD5.

Reviewer #2 (Remarks to the Author):

On the revised manuscript, Chu et al. work to address the concerns previously raised. I commend the authors on the effort to reanalyze the data. I have some comments below.

1 - The usage of harmony yielded a more reasonable cell type distribution, and the plot with marker genes bring more confidence to the cell annotation. This new analysis needs to be described in the methods.

2 - The authors raise in their comments a very likely explanation for the immune overrepresentation in the data. I believe these comments are important for the adequate evaluation of the results by future readers, and should be included in the text. It also seems to be a limitation of the study, particularly given the remark in the discussion about "a shift in the renal cellular landscape".

3 - The code available is unsatisfactory and does not allow for reproducing the results. The code currently available only performs preliminary normalization and clustering. It does not present methods such as harmony, transfer anchors and module scoring. Furthermore, it seems the QC filtering was performed after clustering and normalization. It may not impact the results, since the new clustering were obtained after harmony batch correction, not present in the code.

Reviewer #3 (Remarks to the Author):

The authors have addressed most of the comments raised by the reviewer. However, there is still a need for more data presentation regarding the influence of OTUD5 in the general ferroptosis setting beyond ischemia/reperfusion. As such, a minor study is suggested to fill this gap.

1. In supplementary Fig 4, the authors have provided data on the effect of OTUD5 overexpression in erastin-induced cell death, and the effect of overexpression and knockdown in erastin/RSL3 plus H/R-induced cell death. Unfortunately, due to the additional effect of H/R, it is challenging to evaluate the impact of OTUD5 alone. Therefore, the following minor studies are recommended to assess the role of OTUD5 in ferroptosis regulation:

- i) Evaluate the effect of overexpression of OTUD5 on RSL-induced ferroptosis (without H/R).
- ii) Assess the effect of knockdown (or knockout) of OTUD5 in RSL3-induced ferroptosis (without H/R).
- iii) Assess the effect of knockdown (or knockout) of OTUD5 in erastin-induced ferroptosis (without H/R).

In addition, to provide a clearer explanation of the results, it is suggested that the authors edit the description and presentation of the results as follows:

2. Line 132: "We next confirmed the presence of ferroptosis during AKI, as evidenced by elevated 4-hydroxynonenal (4-HNE), in I/R-treated kidneys (Fig. 1e)."

As previously discussed in the initial review, merely presenting the presence of 4-HNE alone is insufficient to confirm the presence of ferroptosis. To enhance the clarity of the statement, consider revising it as follows or like that: "I/R-treated kidneys exhibited tubular cell death accompanied by lipid peroxidation, as indicated by TUNEL staining and elevated 4-HNE, suggesting the presence of ferroptosis." Additionally, it is recommended to present TUNEL staining data in Fig 1 (or corresponding supplementary figure) in addition to 4-HNE.

3. Line 228: "We found that OTUD5 had slight effects on erastin-induced ferroptosis

(Supplementary Fig. 4a)."

Since there is no significant effect observed, it would be more appropriate to state, "OTUD5 overexpression showed no significant effect." There is no need to overemphasize the significance of OTUD5 in every ferroptosis setting. It is important for future research to honestly report the obtained data and its limitations. This issue applies to comment #1. If OTDU5 is more important in H/R conditions and has a limited effect in other ferroptosis situations, that fact is an important finding that should be described.

Review Comments:

Reviewer #1 (Remarks to the Author):

In response to the comments made by the previous reviewer, the authors completed a large number of supplementary experiments, revised the manuscript in detail, and answered and clarified the questions raised by the reviewer. The authors used single-cell sequencing and spatial transcriptome methods, which showed the unique advantages of this experimental method although the cost was high, and successfully demonstrated the mechanism of GPX-4 mediating ferroptosis by degrading the deubiquitylated protein OTUD5.

R: We are grateful for the reviewer's positive feedback and appreciate the constructive comments provided during the initial revision, which significantly contributed to the advancement of our manuscript. We acknowledge that the use of single-cell sequencing and spatial transcriptome methods indeed comes with a higher cost; however, we believe the unique insights these methods offer, particularly in demonstrating the mechanism by which GPX-4 mediates ferroptosis through the degradation of deubiquitinated OTUD5, justify the investment. We have strived to clearly present these complex methods and their benefits in our revised manuscript, and we hope that our responses and the supplementary experiments adequately address the queries and suggestions raised.

Reviewer #2 (Remarks to the Author):

On the revised manuscript, Chu et al. work to address the concerns previously raised. I commend the authors on the effort to reanalyze the data. I have some comments below.

1 The usage of harmony yielded a more reasonable cell type distribution, and the plot with marker genes bring more confidence to the cell annotation. This new analysis needs to be described in the methods.

R: We appreciate the reviewer's acknowledgment of our efforts in reanalyzing the data. In accordance with your valuable suggestion, we have described the utilization of the 'harmony' package in the methods section of our revised manuscript (page 16).

2 The authors raise in their comments a very likely explanation for the immune overrepresentation in the data. I believe these comments are important for the adequate evaluation of the results by future readers, and should be included in the text. It also seems to be a limitation of the study, particularly given the remark in the discussion about "a shift in the renal cellular landscape".

R: We thank the reviewer for the insightful comments. We have addressed the potential overrepresentation of immune cells in the data within the 'Discussion' section, providing clarification on this point as it relates to the interpretation of the renal

cellular landscape shift mentioned. This revised section now includes commentary on how the selection process during sample preparation may impact cell representation, recognizing it as a study limitation. We hope that these amendments will aid future readers in the thorough evaluation of our results.

3 The code available is unsatisfactory and does not allow for reproducing the results. The code currently available only performs preliminary normalization and clustering. It does not present methods such as harmony, transfer anchors and module scoring. Furthermore, it seems the QC filtering was performed after clustering and normalization. It may not impact the results, since the new clustering were obtained after harmony batch correction, not present in the code.

R: We thank the reviewer for highlighting the deficiencies in the code provided. In response, we have revised and updated our analysis pipeline to include all relevant methods, such as harmony integration, transfer anchors, and module scoring. We have also ensured that the quality control filtering is appropriately positioned in the analytical workflow. The updated and complete code, which allows for full reproduction of the results, is now available on GitHub at <https://github.com/Toby111/scRNA-spatial-code.git>. The specific file “scRNA and ST analysis pipeline_junliu” contains the code.

Reviewer #3 (Remarks to the Author):

The authors have addressed most of the comments raised by the reviewer. However, there is still a need for more data presentation regarding the influence of OTUD5 in the general ferroptosis setting beyond ischemia/reperfusion. As such, a minor study is suggested to fill this gap.

1. In supplementary Fig 4, the authors have provided data on the effect of OTUD5 overexpression in erastin-induced cell death, and the effect of overexpression and knockdown in erastin/RSL3 plus H/R-induced cell death. Unfortunately, due to the additional effect of H/R, it is challenging to evaluate the impact of OTUD5 alone. Therefore, the following minor studies are recommended to assess the role of OTUD5 in ferroptosis regulation:

i) Evaluate the effect of overexpression of OTUD5 on RSL3-induced ferroptosis (without H/R).

ii) Assess the effect of knockdown (or knockout) of OTUD5 in RSL3-induced ferroptosis (without H/R).

iii) Assess the effect of knockdown (or knockout) of OTUD5 in erastin-induced ferroptosis (without H/R).

R: We are grateful for the reviewer's detailed suggestions and have conducted the

recommended additional studies. The results of these experiments have now been incorporated into Supplementary Figure 4a-d. To clarify the impact of OTUD5 on ferroptosis independently of H/R conditions, we have evaluated the effect of OTUD5 overexpression and knockdown (or knockout) on RSL3- and erastin-induced ferroptosis. The manuscript has been updated to include these findings, with the relevant text amended on lines 233-237 (page 6) to reflect this new data and ensure a comprehensive understanding of OTUD5's role in ferroptosis regulation.

In addition, to provide a clearer explanation of the results, it is suggested that the authors edit the description and presentation of the results as follows:

2. Line 132: *"We next confirmed the presence of ferroptosis during AKI, as evidenced by elevated 4-hydroxynonenal (4-HNE), in I/R-treated kidneys (Fig. 1e)."*

As previously discussed in the initial review, merely presenting the presence of 4-HNE alone is insufficient to confirm the presence of ferroptosis. To enhance the clarity of the statement, consider revising it as follows or like that: "I/R-treated kidneys exhibited tubular cell death accompanied by lipid peroxidation, as indicated by TUNEL staining and elevated 4-HNE, suggesting the presence of ferroptosis." Additionally, it is recommended to present TUNEL staining data in Fig 1 (or corresponding supplementary figure) in addition to 4-HNE.

R: We appreciate the reviewer's constructive feedback aimed at enhancing the clarity of our results presentation. In line with the suggestions, we have revised the manuscript text to more accurately reflect the implications of our findings regarding ferroptosis during AKI. The specific passage (now lines 111-113, page 3) has been updated to include a more cautious interpretation of the presence of 4-HNE and its relationship to tubular cell death and lipid peroxidation. Furthermore, we have supplemented our revised manuscript with the TUNEL staining data, which is now presented in Supplementary Figure 1i to support our findings.

3. Line 228: *"We found that OTUD5 had slight effects on erastin-induced ferroptosis (Supplementary Fig. 4a)." Since there is no significant effect observed, it would be more appropriate to state, "OTUD5 overexpression showed no significant effect." There is no need to overemphasize the significance of OTUD5 in every ferroptosis setting. It is important for future research to honestly report the obtained data and its limitations. This issue applies to comment #1. If OTUD5 is more important in H/R conditions and has a limited effect in other ferroptosis situations, that fact is an important finding that should be described.*

R: We are thankful to the reviewer for this critical observation. We recognize the importance of accurately reflecting the impact of OTUD5 on renal tubular cell ferroptosis across different conditions. In accordance with the comments, we have not only conducted additional experiments but have also carefully revised the

manuscript to ensure that our descriptions are in line with the observed data. The phrase in question on line 228 has been amended to state, “altering OTUD5 levels through overexpression or knockdown did not significantly affect ferroptosis triggered by erastin or RSL3,” thereby presenting our findings more honestly and without overemphasis. We have also updated Supplementary Figure 4 and its description in the manuscript to clearly delineate the specific role of OTUD5 in H/R conditions compared to other ferroptosis contexts. We apologize for any misunderstanding in the initial revision and hope that these changes address the reviewer's concerns.

REVIEWERS' COMMENTS

Reviewer #3 (Remarks to the Author):

The authors made appropriate revision to the manuscript with providing additional data. The provided limitaiton showing the specific contribution of OTUD5 in H/R injury is important information.

As a final comment, I would like to request an only minor correction to the legend in Supplementary Figure 4. The title legend "OTUD5 protects renal tubular cells from ferroptosis in response to H/R injury" in supple fig 4 does not fit, since this figure does not show data on H/R injury. Thus, "No significant effect of OTUD5 on ferroptosis induced by erastin and RSL3 in renal tubular cells" or similar would better fit the actual content.

REVIEWERS' COMMENTS

Reviewer #3 (Remarks to the Author):

The authors made appropriate revision to the manuscript with providing additional data. The provided limitaiton showing the specific contribution of OTUD5 in H/R injury is important information.

As a final comment, I would like to request an only minor correction to the legend in Supplementary Figure 4. The title legend "OTUD5 protects renal tubular cells from ferroptosis in response to H/R injury" in supple fig 4 does not fit, since this figure does not show data on H/R injury. Thus, "No significant effect of OTUD5 on ferroptosis induced by erastin and RSL3 in renal tubular cells" or similar would better fit the actual content.

R: We are grateful for the reviewer's positive feedback and appreciate the detailed suggestion. In response, we have revised the title of Supplementary Figure 4 to "OTUD5's Minimal Effect on Erastin and RSL3-Induced Ferroptosis in Renal Tubular Cells".